# VCWorld: A Biological World Model for Virtual Cell Simulation

**Zhijian Wei**[1,2,*]  **Runze Ma**[1,2,*]  **Zichen Wang**[1]  **Zhongmin Li**[1]
**Shuotong Song**[1]  **Shuangjia Zheng**[1,†]

[1] Shanghai Jiao Tong University [2] NeoLife AI
`{realman_01, runze.ma, zichen_w, shuangjia.zheng}@sjtu.edu.cn`

## Abstract

Virtual cell modeling aims to predict cellular responses to perturbations. Existing virtual cell models rely heavily on large-scale single-cell datasets, learning explicit mappings between gene expression and perturbations. Although recent models attempt to incorporate multi-source biological information, their generalization remains constrained by data quality, coverage, and batch effects. More critically, these models often function as black boxes, offering predictions without interpretability or consistency with biological principles, which undermines their credibility in scientific research. To address these challenges, we present VCWorld, a cell-level white-box simulator that integrates structured biological knowledge with the iterative reasoning capabilities of large language models to instantiate a biological world model. VCWorld operates in a data-efficient manner to reproduce perturbation-induced signaling cascades and generates interpretable, stepwise predictions alongside explicit mechanistic hypotheses. In drug perturbation benchmarks, VCWorld achieves state-of-the-art predictive performance, and the inferred mechanistic pathways are consistent with publicly available biological evidence. Our code is publicly available at https://github.com/GENTEL-lab/VCWorld.

## 1 Introduction

Cells, the fundamental units of life, maintain organismal function and homeostasis through a complex interplay of biochemical processes (Alberts et al., 2002). A central challenge in modern biology and drug discovery is to understand and predict how cells respond to external perturbations, such as drug treatments or genetic edits (Liberali et al., 2015; Lotfollahi et al., 2019). The ability to forecast these cellular state changes *in silico* would not only illuminate the mechanisms of complex diseases but also accelerate the development of novel therapeutics by reducing the time and cost of experimental screening (Del Sol et al., 2010). The concept of the virtual cell, which leverages computational models to simulate cellular behavior, has emerged as a promising paradigm to address this challenge (Bunne et al., 2024).

Recent advances in deep learning, coupled with significant progress in single-cell sequencing technologies, have spurred the development of various virtual cell models (Lopez et al., 2018; Lotfollahi et al., 2023; Adduri et al., 2025; Tang et al., 2025; Klein et al., 2025). These models typically learn an end-to-end mapping from a given perturbation to a corresponding gene expression profile, trained on large-scale perturbation-response datasets. However, prevailing approaches suffer from two critical limitations. First, they are heavily reliant on the scale, quality, and coverage of the training data. The data-hungry nature makes these models expensive to train. More importantly, it also limits their ability to generalize to novel perturbations that are not present in the training data (Li et al., 2024; Ahlmann-Eltze et al., 2024). Second, these models operate as black boxes. While they may yield predictive outputs, they fail to provide clear, verifiable mechanistic explanations for predictions (Hassija et al., 2024; Noutahi et al., 2025). This lack of interpretability severely undermines their trustworthiness and utility in scientific discovery, making them difficult for biologists to rely upon for designing downstream experiments.

---

*These authors contributed equally to this work. † Corresponding author.

We argue that an ideal virtual cell model should not only provide accurate predictions but also be data-efficient, interpretable, and aligned with established biological principles. Rather than relying solely on statistical correlations, it ought to integrate fundamental biological knowledge to capture and anticipate cellular responses. Moreover, its reasoning process should be transparent, explicitly revealing the causal mechanisms that underlie predictions and grounding them in the well-established frameworks of cell signaling and gene regulation.

To overcome the above challenges, we introduce VCWorld, a cell-level white-box simulator(Figure 1). The core of VCWorld is a biological world model that simulates the dynamic response of a cell to drug perturbations. Instead of relying solely on statistical patterns, VCWorld integrates structured biological knowledge, such as signaling pathways, protein-protein interactions, and gene regulatory networks, with the iterative reasoning capabilities of Large Language Models (LLMs) (Dubey et al., 2024; Guo et al., 2025). This design allows the model to generalize from limited training data by leveraging a vast repository of open-world biological knowledge. Crucially, VCWorld generates a transparent, traceable reasoning path for each prediction, offering a step-by-step mechanistic explanation that culminates in verifiable hypotheses. Furthermore, to facilitate finer-grained modeling, we introduce GeneTAK, a new benchmark derived from the large-scale Tahoe-100M dataset (Zhang et al., 2025). GeneTAK reframes cell-drug observations into gene-centric perturbation responses, mitigating data sparsity and enabling models to focus directly on the nuanced impact of a drug on individual genes. We highlight three main contributions of our study:

- We propose VCWorld, a novel cell-level white-box simulator architected as a biological world model. It combines structured biological knowledge with LLM-based reasoning, demonstrating a superior balance of data efficiency, interpretability, and predictive accuracy that overcomes the limitations of existing black-box models.

- We construct and introduce GeneTAK, a new benchmark that transforms cell-drug perturbation data into single-gene response profiles. This allows for more granular and robust modeling of drug effects.

- We demonstrate that VCWorld achieves state-of-the-art performance on both the differential expression (DE) and directional (DIR) prediction tasks on the GeneTAK benchmark, validating the effectiveness of our approach.

## 2 RELATED WORK

### 2.1 VIRTUAL CELL

Virtual cell was originally a computational tool for simulating intracellular biochemical reactions, diffusion, membrane transport, and electro physiological processes (Loew & Schaff, 2001). With the development of AI, the concept has evolved into predictive cell models that integrate large-scale multi-modal biological data to forecast, explain, and guide experimental hypotheses (Bunne et al., 2024; Noutahi et al., 2025). Existing research can be broadly categorized into three approaches: data-driven methods combined with prior knowledge, such as GenePT (Chen & Zou, 2024) and GEARS (Roohani et al., 2024); large-scale pretrained foundational models, like scFoundation (Hao et al., 2024) and scGPT (Cui et al., 2024), trained on tens of millions of cells to learn general representations and excel in perturbation prediction; and generative modeling approaches, including CPA (Lotfollahi et al., 2023), STATE (Adduri et al., 2025), and CellFlow (Klein et al., 2025), which capture complex perturbation effects through decoupled latent spaces, state transitions, or flow-based generation. Further efforts, such as CellForge (Tang et al., 2025), aim to automate data analysis, literature review, and model design, pushing virtual cell toward self-directed evolution. Unlike previous approaches that mainly focus on end-to-end black-box models for cell perturbation prediction, our approach leverages a language model with reasoning and retrieval capabilities to construct a framework that emphasizes interpretability while also achieving superior performance.

### 2.2 LLMS FOR BIOLOGICAL REASONING AND PREDICTION

Recent research focus on leveraging Large Language Models (LLMs) for interpretable biological reasoning, aiming to move beyond traditional black-box predictions. A core approach in this area is the application of Chain-of-Thought (CoT) prompting (Wei et al., 2022) to integrate multi-

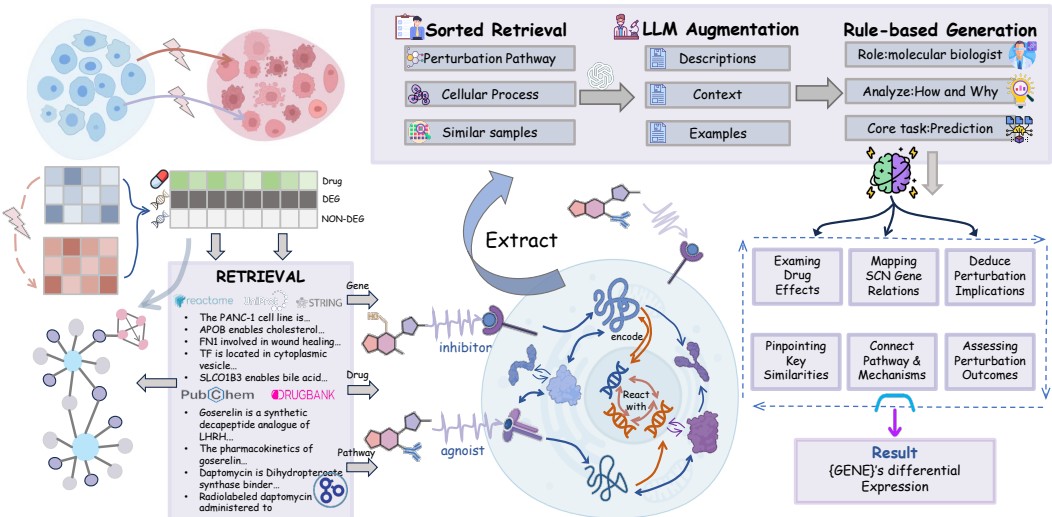

Figure 1: The pipeline of VCWorld. For a given query, VCWorld first retrieves multi-modal biological context from its integrated knowledge base, including pathway information, drug properties, and similar experimental samples. This context is then used to prompt a LLM, which is used to analyze mechanisms and infer how a specific gene will respond. The final output is a prediction for tasks such as differential expression and directional change.

source heterogeneous data. For instance, frameworks like CoTox (Park et al., 2025) and DrugReasoner (Ghaffarzadeh-Esfahani et al., 2025) combine chemical structures, biological pathways, and Gene Ontology (GO) terms to generate interpretable predictions for drug toxicity and approval processes. This paradigm has also been extended to fundamental biological contexts, such as the SUMMER framework (Wu et al., 2025), which utilizes retrieval-augmented generation (RAG) to predict perturbation experiments under gene edits. To further enhance the reasoning capabilities of these models, researchers are actively exploring more advanced training methodologies. These studies (Istrate et al., 2025; Hasanaj et al., 2025) collectively demonstrate the strong potential of CoT to effectively integrate multimodal biological data within the virtual cell domain, enabling more nuanced and interpretable predictions compared to foundational models.

## 2.3 PERT-SEQ DATA REPRESENTATION

Large-scale perturbation datasets (Wu et al., 2024) are foundational for building predictive models of cellular behavior. Recently, scientists have gradually expanded the scope of perturbations. Replogle et al. (2022) enabled predictions of genetic perturbation effects on core cellular functions. Nadig et al. (2024) first introduced the dose effect in the perturbation. Jiang et al. (2025) aided inference of pathway signatures for perturbation-driven changes in disease scenarios. More recently, Wu et al. (2025) released PerturBase, a dedicated database for single-cell perturbation sequencing data, integrating 122 datasets from 46 studies across genetic and chemical perturbations. A landmark in chemical perturbations is Tahoe-100M (Zhang et al., 2025), empowering AI models to predict context-dependent responses to small-molecule drugs across vast cellular diversity. Given the large size and coverage of the Tahoe-100M dataset, we use it as the primary dataset.

## 3 METHOD

### 3.1 TASK FORMULATION

The task of predicting single-cell gene expression responses to perturbations is conventionally defined as a regression problem. Given a dataset $D = \{(x_i, p_i, y_i)\}_{i=1}^{N}$, where $x_i, y_i \in \mathbb{R}^d$ are the pre- and post-perturbation expression profiles and $p_i$ is the perturbation, the goal is to learn a mapping function

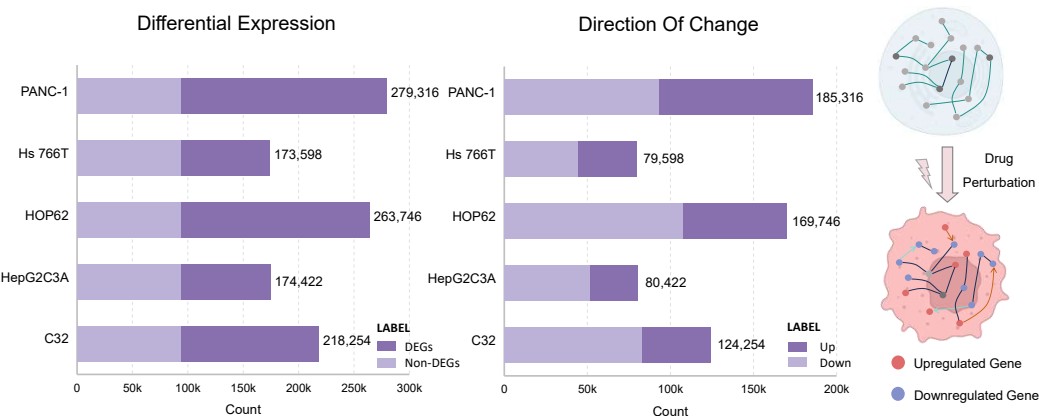

Figure 2: (a) Number of single-cell profiles per cell line. (b) Label distributions for the Differential Expression (DE) and Directional Change (DIR) tasks. (c) Schematic for identifying highly variable genes by comparing perturbed cells against a control group.

$f : \mathbb{R}^d \times \mathcal{P} \to \mathbb{R}^d$. However, the high dimensionality and sparsity of the data pose significant challenges to achieving accurate predictions at the individual gene level. To this end, we reformulate the problem as a *gene-centric classification task*. The fundamental predictive unit is a triplet $(c, p, g)$, which queries the response of a specific gene $g$ to a perturbation $p$ in a given cell lineage $c$. For each such triplet, we derive a corresponding binary label, let's denote it as $l \in \{0, 1\}$, which represents the outcome for our two fine-grained classification tasks: (i) **Differential Expression (DE):** $l = 1$ if gene $g$ is differentially expressed, and $l = 0$ otherwise. (ii) **Directional Change (DIR):** $l = 1$ for upregulation and $l = 0$ for downregulation. This transformation converts the original dataset $D$ into a new, larger corpus of labeled triplets suitable for training a classifier.

These tasks can be addressed by a new paradigm that shifts from numerical regression to knowledge-based inference. The approach utilizes a reasoning engine, such as a Large Language Model (LLM) $L$, to process not only the query but also a rich set of biological context (BioContext). This context is dynamically retrieved from external knowledge bases (*e.g.*, signaling pathways, protein-protein interactions). The problem is then transformed into a text-based inference function as follows: $f : L(c, p, g, \texttt{BioContext}) \to \texttt{Prediction}$.

## 3.2 GENETAK

To facilitate the comprehensive and fair assessment of large language models for perturbation prediction, we present GeneTAK, a novel benchmark dataset. GeneTAK is curated from the Tahoe-100M raw expression matrix and comprises 5 distinct cell lines across 348 drug perturbations, designed explicitly to evaluate the generalization capabilities of models in this domain.

**Dataset Curation.** The GeneTAK expression matrix was first restricted to the 2,000 most highly variable genes, aligning it with the common input dimensions of generative perturbation prediction models. Subsequently, to ensure a diverse and representative selection of cell lines, we performed a Principal Component Analysis (PCA) visualization (Greenacre et al., 2022) (see Appendix A). Guided by this analysis, we selected a final set of five cell lines that includes both a main cluster and several distinct outliers: C32, HOP62, HepG2C3A, Hs 766T, and PANC-1. Notably, this selection is consistent with the cell lines used in the work of STATE (Adduri et al., 2025), ensuring a degree of comparability and fairness with previous benchmarks.

Then we generated labels for perturbation-gene pairs, denoted as $(p, g)$. The labels were determined by identifying differentially expressed genes (DEGs) for each perturbation using the Wilcoxon signed-rank test (Woolson, 2007). To ensure label quality, we applied strict criteria based on a significant adjusted p-value and consistent expression changes across biological replicates. The resulting dataset of labeled pairs was then split by perturbation into training and test sets at a 3:7 ratio. This specific ratio was intentionally chosen to simulate a challenging few-shot learning scenario, consistent with

the evaluation framework used by Adduri et al. (2025) and allowing for a direct comparison of model performance under low-data conditions. The split maintains a similar distribution of DEGs per perturbation in both sets. The final label distribution is shown in Figure 2, with a complete description of all data processing methods provided in Appendix A.

## 3.3 VCWORLD

Our proposed framework, VCWorld (Figure 1), transforms biological prediction into a multi-stage reasoning process powered by a Large Language Model (LLM). It consists of three key stages: (1) generating rich, symbolic representations for all biological entities; (2) retrieving a support set of causal evidence guided by open-world biological knowledge; and (3) synthesizing all information through a Chain-of-Thought process to produce an interpretable prediction.

## 3.4 CONSTRUCTION OF THE OPEN-WORLD BIOLOGICAL KNOWLEDGE GRAPH

To construct the open-world biological knowledge graph, we integrated several authoritative databases: PubChem, which provides large-scale compound structures and bioactivity data (Kim et al., 2023); DrugBank, which includes chemical, pharmacological, and clinical annotations of drugs and their target associations (Knox et al., 2024); UniProt, which offers comprehensive protein sequences and functional annotations (Consortium, 2019); Gene Ontology (GO), which defines a unified ontology across molecular function, biological process, and cellular component (Ashburner et al., 2000); Reactome, which curates systematically organized molecular reactions and pathways (Fabregat et al., 2018); STRING, which compiles protein–protein interactions from experiments and predictions (Szklarczyk et al., 2021); and CORUM, which catalogs experimentally validated mammalian protein complexes (Ruepp et al., 2007). Together, these databases cover multi-scale biological knowledge from compounds and drugs to genes, proteins, pathways, and complexes, ensuring the comprehensiveness of the graph.

In the knowledge graph construction process, compounds, drugs, genes, proteins, pathways, and complexes were represented as entity nodes, while interactions, annotations, and hierarchical relationships were encoded as heterogeneous edges. Cross-database integration was achieved through standardized identifiers (*e.g.*, InChIKey, UniProt ID, GO terms), followed by redundancy removal and conflict resolution. The resulting graph systematically represents biomolecules and their interactions, providing a foundation for cellular perturbation prediction.

### 3.4.1 GENERATIVE NODE FEATURE REPRESENTATION VIA LLMS

Let the open-world biological knowledge be denoted as $G = (\mathcal{V}, \mathcal{E}, \mathcal{R})$, where $\mathcal{V}$ is the set of biological entities (nodes), $\mathcal{R}$ is the set of relation types, and $\mathcal{E} \subseteq \mathcal{V} \times \mathcal{R} \times \mathcal{V}$ is the set of triples or edges. The first stage of our framework focuses on defining each biological entity $v \in \mathcal{V}$. To move beyond simple numerical vectors, we generate a rich symbolic representation for each node. For each node $v$, we first extract its local neighborhood subgraph, $N_k(v)$, defined as the set of all triples within a $k$-hop distance from $v$. We then construct a structured prompt $P_v$, using a template function $f_{\text{prompt}}$:

$$P_v = f_{\text{prompt}}(v, N_k(v)). \tag{1}$$

This function serializes the node's core attributes and its neighborhood triples into a natural language query. Subsequently, our framework employs an LLM to function as a feature generator. The LLM $L$ then processes this prompt to generate a comprehensive textual description(see Appendix D), $d_v$, serving as the node's initial feature representation:

$$d_v = L(P_v) \tag{2}$$

This process yields a context-aware representation $d_v \in \mathcal{T}$ (where $\mathcal{T}$ is the space of all possible texts) that preserves biological semantics, offering a more expressive alternative to static embeddings.

### 3.4.2 GRAPH-GUIDED CAUSAL EVIDENCE FRAMEWORK

With rich node representations established, the second stage retrieves relevant experimental cases to form a basis for reasoning. Our training corpus $\mathcal{D}$, consists of $M$ labeled instances derived from the original data, can be formulated as $\mathcal{D} = \{(q_i, l_i)\}_{i=1}^{M}$. Here, each query $q_i$ is a triplet

$(c_i, p_i, g_i)$, and $l_i$ is its associated ground-truth binary label ($l_i \in \{0, 1\}$). Given a new query $q_{input} = (c_{\text{input}}, p_{\text{input}}, g_{\text{input}})$, our goal is to construct an evidence support set $S(q_{\text{input}}) \subset \mathcal{D}$.

To ground the LLM's prediction in empirical data, we introduce a structured retrieval mechanism that goes beyond standard Retrieval-Augmented Generation (RAG) (Lewis et al., 2020). The core of this mechanism is to identify and retrieve the most relevant historical cases from our training corpus $\mathcal{D}$ for given input query $q_{\text{input}}$. To quantify this relevance, we score the similarity between the input query $q_{\text{input}}$ and each query $q_i \in \mathcal{D}$. Instead of relying solely on semantic similarity, our method computes a hybrid similarity score, $\text{Sim}(q_{\text{input}}, q_i)$, that also leverages knowledge-graph topology information:

$$\text{Sim}(q_{\text{input}}, q_i) = \alpha \cdot \text{Sim}_{\text{sem}}(d_{q_{\text{input}}}, d_{q_i}) + (1 - \alpha) \cdot \text{Sim}_{\text{struct}}(q_{\text{input}}, q_i) \tag{3}$$

where $\text{Sim}_{\text{sem}}(\cdot)$ is the semantic similarity, calculated as the cosine similarity between the LLM-generated feature descriptions (from Eq. 2), and $\text{Sim}_{\text{struct}}(\cdot)$ is a graph-based structural similarity metric (*i.e.*, path-based similarity). The hyperparameter $\alpha \in [0, 1]$ balances the contribution of the semantic and structural components.

This score allows us to assemble a multifaceted evidence set. Instead of retrieving a single list of similar items, we retrieve two disjoint subsets by searching within predefined outcome groups as follows:

**Analogue Cases ($S_{\text{analog}}$):** The top-$k_a$ instances from the subset of $\mathcal{D}$ with a positive outcome (*i.e.*, label $l = 1$), ranked by their similarity to the input query.

$$S_{\text{analog}}(q_{\text{input}}) = \underset{q_i \in \{q \in \mathcal{D} \,|\, l=1\}}{\arg \text{top-}k_a} \text{Sim}(q_{\text{input}}, q_i) \tag{4}$$

**Contrast Cases ($S_{\text{contrast}}$):** The top-$k_c$ instances from the subset of $\mathcal{D}$ with a negative outcome (i.e., label $l = 0$), similarly ranked by similarity.

$$S_{\text{contrast}}(q_{\text{input}}) = \underset{q_i \in \{q \in \mathcal{D} \,|\, l=0\}}{\arg \text{top-}k_c} \text{Sim}(q_{\text{input}}, q_i) \tag{5}$$

The final evidence support set is the union: $S(q_{\text{input}}) = S_{\text{analog}} \cup S_{\text{contrast}}$. This curated set provides a balanced, contextual foundation upon which the final reasoning process is built.

### 3.4.3 EVIDENCE SYNTHESIS CHAIN-OF-THOUGHT REASONING

In the final stage, the LLM acts as a computational biologist, which we formulate as a Chain-of-Thought (CoT) reasoning task. A final prompt, $P_{\text{CoT}}$, is synthesized from the symbolic representation of the query, $d_{q_{\text{input}}}$, and the retrieved evidence set, $S(q_{\text{input}})$:

$$P_{\text{CoT}} = f_{\text{CoT\_prompt}}(d_{q_{\text{input}}}, S_{\text{analog}}, S_{\text{contrast}}) \tag{6}$$

The function $f_{\text{CoT\_prompt}}$ formats these inputs using a structured template that first presents the biological query, followed by the lists of analogue and comparison cases as evidence. Finally, it instructs the LLM to provide step-by-step reasoning. (A detailed example is in Appendix D).

The LLM then processes this prompt to generate a text string, $O_{\text{final}}$, we then use a parsing function, $f_{\text{parse}}$, to extract the structured prediction $\hat{l}$ and the textual explanation $E$ from the output:

$$O_{\text{final}} = L(P_{\text{CoT}}) \tag{7}$$

$$(\hat{l}, E) = f_{\text{parse}}(O_{\text{final}}) \tag{8}$$

This process compels the LLM to explicitly integrate qualitative knowledge ($d_{q_{\text{input}}}$) with empirical evidence ($S(q_{\text{input}})$), producing a self-validating and fully interpretable output. In this work, we use Gemeni2.5-Flash (Comanici et al., 2025) as our reasoning model.

## 4 EXPERIMENT

### 4.1 BASELINE

We benchmarked performance against several baselines using GeneTAK, including RANDOM, which is a naive baseline that is assumed to have a statistically accurate prediction of $50\%$ for any

Table 1: Overall accuracy on DE and DIR tasks. The best results are shown in bold, and the second-best results are shown with underlines.

| Task | Model | C32 | HepG2C3A | HOP62 | Hs 766T | PANC-1 |
|------|-------|-----|----------|-------|---------|--------|
| **DE** | RANDOM | $0.50 \pm .00$ | $0.50 \pm .00$ | $0.50 \pm .00$ | $0.50 \pm .00$ | $0.50 \pm .00$ |
| | GAT | $0.62 \pm .03$ | $0.54 \pm .02$ | $0.64 \pm .04$ | $0.57 \pm .01$ | $0.58 \pm .01$ |
| | CPA | $0.17 \pm .04$ | $0.18 \pm .02$ | $0.21 \pm .03$ | $0.30 \pm .04$ | $0.17 \pm .02$ |
| | scVI | $\underline{0.66} \pm .02$ | $0.48 \pm .01$ | $0.64 \pm .02$ | $0.61 \pm .01$ | $\underline{0.68} \pm .02$ |
| | STATE | $0.17 \pm .02$ | $0.38 \pm .01$ | $0.41 \pm .01$ | $0.09 \pm .02$ | $0.47 \pm .01$ |
| | **VCWorld** (Retrieval-only) | $0.57 \pm .01$ | $0.53 \pm .00$ | $0.58 \pm .01$ | $0.57 \pm .02$ | $0.58 \pm .01$ |
| | **VCWorld** (w/o Biocontext) | $0.52 \pm .03$ | $0.54 \pm .03$ | $0.51 \pm .02$ | $0.52 \pm .01$ | $0.50 \pm .03$ |
| | **VCWorld** (w/o CoT) | $0.61 \pm .03$ | $0.57 \pm .00$ | $0.59 \pm .01$ | $0.60 \pm .02$ | $0.56 \pm .03$ |
| | **VCWorld** (Llama3-8B) | $0.35 \pm .01$ | $0.36 \pm .00$ | $0.41 \pm .01$ | $0.36 \pm .00$ | $0.39 \pm .02$ |
| | **VCWorld** (Qwen3-4B) | $0.41 \pm .00$ | $0.47 \pm .01$ | $0.46 \pm .00$ | $0.49 \pm .02$ | $0.46 \pm .01$ |
| | **VCWorld** (Qwen2.5-7B) | $0.54 \pm .01$ | $0.54 \pm .00$ | $0.55 \pm .00$ | $0.58 \pm .01$ | $0.64 \pm .02$ |
| | **VCWorld** (Qwen2.5-14B) | $0.65 \pm .02$ | $\underline{0.66} \pm .01$ | $\mathbf{0.72} \pm .01$ | $\underline{0.67} \pm .00$ | $\mathbf{0.76} \pm .01$ |
| | **VCWorld** (Gemini-2.5-Flash) | $\mathbf{0.70} \pm .00$ | $\mathbf{0.68} \pm .00$ | $\underline{0.71} \pm .02$ | $\mathbf{0.68} \pm .00$ | $0.61 \pm .02$ |
| **DIR** | RANDOM | $0.50 \pm .00$ | $0.50 \pm .00$ | $0.50 \pm .00$ | $0.50 \pm .00$ | $0.50 \pm .00$ |
| | GAT | $0.58 \pm .03$ | $0.61 \pm .01$ | $0.54 \pm .02$ | $0.55 \pm .01$ | $0.52 \pm .01$ |
| | CPA | $0.22 \pm .01$ | $0.19 \pm .03$ | $0.17 \pm .01$ | $0.21 \pm .02$ | $0.15 \pm .01$ |
| | scVI | $0.47 \pm .01$ | $0.46 \pm .01$ | $0.41 \pm .03$ | $0.40 \pm .01$ | $0.38 \pm .02$ |
| | STATE | $0.49 \pm .01$ | $0.51 \pm .03$ | $0.50 \pm .01$ | $0.55 \pm .02$ | $0.51 \pm .01$ |
| | **VCWorld** (Retrieval-only) | $0.47 \pm .01$ | $0.47 \pm .02$ | $0.47 \pm .01$ | $0.50 \pm .00$ | $0.51 \pm .01$ |
| | **VCWorld** (w/o Biocontext) | $0.37 \pm .00$ | $0.36 \pm .01$ | $0.36 \pm .01$ | $0.17 \pm .03$ | $0.34 \pm .02$ |
| | **VCWorld** (w/o CoT) | $0.42 \pm .01$ | $0.45 \pm .00$ | $0.51 \pm .01$ | $0.47 \pm .01$ | $0.45 \pm .00$ |
| | **VCWorld** (Llama3-8B) | $0.37 \pm .01$ | $0.34 \pm .02$ | $0.38 \pm .04$ | $0.40 \pm .01$ | $0.35 \pm .01$ |
| | **VCWorld** (Qwen3-4B) | $0.54 \pm .01$ | $0.55 \pm .00$ | $0.52 \pm .00$ | $0.48 \pm .02$ | $0.50 \pm .01$ |
| | **VCWorld** (Qwen2.5-7B) | $0.56 \pm .02$ | $0.56 \pm .01$ | $0.55 \pm .00$ | $0.51 \pm .00$ | $0.53 \pm .01$ |
| | **VCWorld** (Qwen2.5-14B) | $\underline{0.67} \pm .01$ | $\underline{0.59} \pm .00$ | $\underline{0.65} \pm .01$ | $\mathbf{0.71} \pm .00$ | $\underline{0.66} \pm .01$ |
| | **VCWorld** (Gemini-2.5-Flash) | $\mathbf{0.72} \pm 0.01$ | $\mathbf{0.68} \pm .00$ | $\mathbf{0.67} \pm .00$ | $\underline{0.66} \pm .00$ | $\mathbf{0.69} \pm .00$ |

binary classification task without any training or prior knowledge; three published deep learning models: scVI (Lopez et al., 2018) is a conditional variational self-encoder that conditionally models perturbation effects in potential space, CPA (Lotfollahi et al., 2023) recognizes novel perturbations by learning disentangled and linearly combined latent embeddings, STATE (Adduri et al., 2025), the current SOTA model that demonstrates the best overall performance in the drug perturbation prediction task.

## 4.2 METRICS

We evaluate model performance on two primary aspects: differentially expressed gene (DEG) prediction performance and LLM reasoning robustness. For a fair comparison, predictions from baseline models that output continuous expression values (*e.g.*, STATE, scVI) are first converted into binary value. This is achieved by applying the Wilcoxon Signed-Rank Test (Woolson, 2007) to the predicted profiles, mirroring the ground-truth label generation process. All metrics below are calculated on these standardized outputs.

**Metric for DEG Prediction Performance.** We evaluate DEG prediction performance using four standard classification metrics: accuracy, precision, recall, and F1 score. Accuracy provides an overall measure of the proportion of correctly classified genes, but because most genes are not differentially expressed (resulting in substantial class imbalance), Accuracy alone can be misleading. To obtain a more informative assessment of performance on the positive class (DEGs), we therefore also report Precision, which reflects the ability to avoid false positives, and Recall, which reflects the ability to avoid false negatives. The F1 score, defined as the harmonic mean of Precision and Recall, provides a single, balanced summary of these two aspects. In addition, to avoid bias toward models trained with regression-style objectives, we also compute the AUROC and AUPRC.

Table 2: Performance comparison of different models on DE and DIR tasks.

| Task | Model | Metric | C32 | HepG2C3A | HOP62 | Hs 766T | PANC-1 | Average |
|------|-------|--------|-----|----------|-------|---------|--------|---------|
| DE | GAT | Precision | 0.63 | 0.56 | 0.58 | 0.50 | 0.63 | 0.58 |
| | | Recall | 0.42 | 0.43 | 0.46 | 0.39 | 0.47 | 0.43 |
| | | F1-Score | 0.50 | 0.49 | 0.51 | 0.44 | 0.54 | 0.49 |
| | STATE | Precision | 0.12 | 0.10 | 0.18 | 0.10 | 0.19 | 0.13 |
| | | Recall | 0.15 | 0.32 | 0.36 | 0.09 | 0.37 | 0.26 |
| | | F1-Score | 0.14 | 0.14 | 0.24 | 0.10 | 0.25 | 0.17 |
| | CPA | Precision | 0.11 | 0.10 | 0.14 | 0.07 | 0.17 | 0.12 |
| | | Recall | 0.02 | 0.02 | 0.02 | 0.01 | 0.02 | 0.02 |
| | | F1-Score | 0.03 | 0.03 | 0.04 | 0.02 | 0.04 | 0.03 |
| | scVI | Precision | 0.13 | 0.09 | 0.18 | 0.08 | 0.19 | 0.13 |
| | | Recall | 0.48 | 0.39 | 0.52 | 0.38 | 0.51 | 0.46 |
| | | F1-Score | 0.21 | 0.14 | 0.27 | 0.14 | 0.28 | 0.21 |
| | VCWorld | Precision | 0.60 | 0.64 | 0.61 | 0.57 | 0.52 | 0.59 |
| | | Recall | 0.70 | 0.68 | 0.71 | 0.68 | 0.61 | 0.68 |
| | | F1-Score | **0.65** | **0.66** | **0.66** | **0.62** | **0.56** | **0.63** |
| DIR | GAT | Precision | 0.62 | 0.54 | 0.64 | 0.57 | 0.58 | 0.59 |
| | | Recall | 0.37 | 0.39 | 0.49 | 0.44 | 0.52 | 0.44 |
| | | F1-Score | 0.46 | 0.45 | 0.56 | 0.50 | 0.55 | 0.50 |
| | STATE | Precision | 0.14 | 0.18 | 0.17 | 0.09 | 0.10 | 0.14 |
| | | Recall | 0.18 | 0.35 | 0.26 | 0.16 | 0.25 | 0.24 |
| | | F1-Score | 0.16 | 0.24 | 0.21 | 0.12 | 0.15 | 0.18 |
| | CPA | Precision | 0.10 | 0.10 | 0.15 | 0.10 | 0.03 | 0.10 |
| | | Recall | 0.02 | 0.02 | 0.02 | 0.03 | 0.19 | 0.06 |
| | | F1-Score | 0.03 | 0.03 | 0.04 | 0.05 | 0.05 | 0.04 |
| | scVI | Precision | 0.08 | 0.08 | 0.10 | 0.11 | 0.09 | 0.09 |
| | | Recall | 0.49 | 0.46 | 0.46 | 0.51 | 0.46 | 0.48 |
| | | F1-Score | 0.14 | 0.14 | 0.17 | 0.18 | 0.15 | 0.16 |
| | VCWorld | Precision | 0.61 | 0.54 | 0.58 | 0.58 | 0.60 | 0.58 |
| | | Recall | 0.72 | 0.68 | 0.67 | 0.66 | 0.69 | 0.68 |
| | | F1-Score | **0.66** | **0.60** | **0.62** | **0.62** | **0.64** | **0.63** |

**Metric for LLM Reasoning Robustness.** We define Q-score to measure the robustness of the LLM's reasoning process. It is the frequency with which the model abstains from making a prediction for a given query (*e.g.*, by stating it has insufficient information). A lower rate signifies higher robustness. The detailed formulas for all metrics are provided in Appendix B.1.

## 4.3 GENE-LEVEL PERFORMANCE

We consider the ability to capture gene-level expression changes as the most fundamental evaluation criterion. To systematically assess this, the task is stratified into two components: Differential Expression (DE) prediction, which identifies whether a gene's expression changes significantly; and Directional Change (DIR) prediction, which determines whether the expression is up-regulated or down-regulated. We use Accuracy to evaluate the models' performance in correctly predicting the ground-truth labels. The results are detailed in Table 1.

**Differential Expression Task Performance.** As shown in Table 1, our model, VCWorld, achieves the highest overall accuracy, frequently exceeding 0.70 across multiple cell lines. Among the baseline models, performance is highly polarized. scVI and GAT demonstrate strong capabilities in detecting gene changes, showing competitive performance that even rivals some LLM variants. This indicates that these traditional models are effective at recognizing that a perturbation has occurred. Conversely,

Table 3: Comparison of AUROC and AUPRC on Differential Expression (DE) prediction.

| Task | Metric | Model | C32 | HepG2C3A | HOP62 | Hs766T | PANC-1 |
|------|--------|-------|-----|----------|-------|--------|--------|
| DE | AUROC | RANDOM | 0.50 | 0.50 | 0.50 | 0.50 | 0.50 |
| | | GAT | **0.65** | **0.59** | **0.69** | **0.62** | **0.68** |
| | | CPA | 0.20 | 0.17 | 0.15 | 0.18 | 0.14 |
| | | scVI | 0.61 | 0.47 | 0.68 | 0.60 | 0.54 |
| | | STATE | 0.62 | 0.49 | 0.63 | 0.57 | 0.58 |
| | | **VCWorld** (Llama3-8B) | 0.32 | 0.34 | 0.28 | 0.40 | 0.33 |
| | | **VCWorld** (Qwen2.5-7B) | 0.45 | 0.51 | 0.56 | 0.51 | 0.53 |
| | | **VCWorld** (Qwen3-4B) | 0.35 | 0.42 | 0.35 | 0.34 | 0.36 |
| | | **VCWorld** (Qwen2.5-14B) | 0.42 | 0.40 | 0.38 | 0.56 | 0.48 |
| | | **VCWorld** (Gemini-2.5-Flash) | 0.51 | 0.42 | 0.47 | 0.46 | 0.52 |
| | AUPRC | RANDOM | 0.61 | 0.58 | 0.68 | 0.73 | 0.78 |
| | | GAT | 0.43 | 0.38 | 0.47 | 0.42 | 0.46 |
| | | CPA | 0.21 | 0.32 | 0.24 | 0.22 | 0.19 |
| | | scVI | 0.42 | 0.48 | 0.52 | 0.56 | 0.65 |
| | | STATE | 0.37 | 0.42 | 0.44 | 0.36 | 0.44 |
| | | **VCWorld** (Llama3-8B) | 0.62 | 0.58 | 0.49 | 0.57 | 0.52 |
| | | **VCWorld** (Qwen2.5-7B) | 0.80 | 0.60 | 0.83 | 0.82 | **0.88** |
| | | **VCWorld** (Qwen3-4B) | 0.76 | 0.54 | 0.76 | 0.72 | 0.76 |
| | | **VCWorld** (Qwen2.5-14B) | **0.85** | 0.71 | 0.82 | **0.89** | 0.87 |
| | | **VCWorld** (Gemini-2.5-Flash) | 0.83 | **0.72** | **0.84** | 0.81 | 0.84 |

CPA and STATE perform poorly on this task, with accuracies falling far below the random baseline of 0.50, suggesting that these models fail to capture the basic perturbation patterns in this specific experimental setup, likely producing highly skewed or incorrect predictions.

**Directional Change Task Performance.** The DIR task is inherently more challenging as it requires inferring the direction of expression changes. Most traditional baselines suffer a significant accuracy drop from the DE task, exposing their inability to determine regulatory directions. The only exception is GAT (0.52–0.61), indicating that modeling gene relationships via graphs provides some directional signals. Conversely, VCWorld achieves stable and superior accuracies (0.65–0.72). By leveraging the biological knowledge and reasoning of LLMs, VCWorld overcomes traditional limitations, accurately inferring not just that a gene is perturbed, but exactly how it is regulated.

## 4.4 CELL-LEVEL AND POPULATION-LEVEL ANALYSIS

We believe that examining the collective response of gene populations offers a more accurate representation of cellular changes. To evaluate this, we quantified the number of DEGs predicted across the GeneTAK dataset (see Appendix B).

The results show clear patterns. scVI predicts 3–5 times more DEGs than the ground truth. STATE is less stable, but on average predicts about twice as many DEGs. These results indicate that both models amplify perturbation effects and classify genes as DEGs too aggressively. This explains their high Accuracy but low Precision, Recall, and F1 Score, which limits their reliability. In contrast, CPA consistently underestimates DEG counts, reflecting a limited ability to capture perturbation effects.

To obtain a more balanced view, we further computed Precision, Recall, F1 score (Table 2), and the threshold-free metrics AUROC , AUPRC (Table 3). The results confirm the above observations: (i) VCWorld achieves the best overall performance, with high and balanced precision (0.59) and recall (0.68), resulting in the highest F1 score (0.63); (ii) scVI attains relatively high recall (0.46) but low precision (0.13), yet still stands out as the strongest baseline; (iii) STATE shows mediocre performance across all metrics, with a low overall F1 score of only 0.17; and (iv) both STATE and scVI amplify perturbation effects, with scVI predicting several times more DEGs than the ground truth and STATE predicting about twice as many. The threshold-free evaluation further sharpens these conclusions: although baselines such as GAT and scVI reach moderate AUROC values (approximately 0.60–0.65), AUROC can be overly optimistic in highly imbalanced settings like gene perturbation, where non-DEGs vastly outnumber DEGs. In contrast, AUPRC, which focuses on

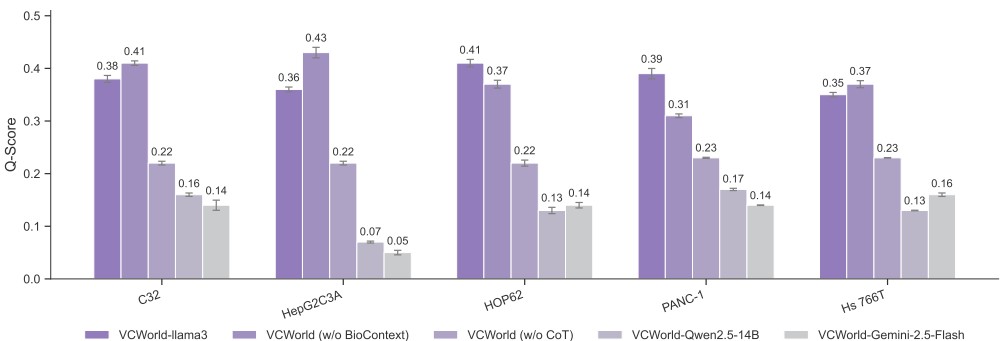

Figure 3: The ablation experiments of VCWorld.

the positive class (perturbed genes), provides a more decisive assessment; here, VCWorld clearly dominates, achieving an average AUPRC above 0.80 (*e.g.*, 0.85 on C32), while the baselines typically remain in the 0.40–0.50 range.

These findings suggest that deep learning models often lack explicit constraints to regulate DEG counts. They fail to capture the true intensity and scope of perturbation effects. In contrast, the design of VCWorld directly addresses these limitations, which explains its superior performance.

### 4.5 ANALYSIS AND ABLATION STUDY

To evaluate the model's tendency to avoid answering questions, we introduce **Q-score** as a dedicated metric for assessing LLM performance. Q-score is formally defined as a measure of the proportion of questions that are not answered normally, reflecting the model's avoidance behavior when faced with complex or sensitive queries. Further details on the computation and usage of Q-score are provided in Appendix B.1.4. Figure 3 reports the Q-scores after ablating different components of the model. The ablation results reveal three key mechanisms driving VCWorld's success:

- **LLM Reasoning Capability is Critical:** Performance scales with the intelligence of the backbone model. We observe a clear trajectory from Llama3-8B (0.37) → Qwen2.5-14B (0.65) → Gemini-2.5-Flash (0.70) on the C32 cell line. This 84% performance leap (Llama3-8B vs. Gemini-2.5-Flash) confirms that the task requires advanced reasoning, not just pattern recognition.
- **BioContext as a Cornerstone:** Removing biological context (VCWorld w/o BioContext) causes performance to plummet to near-random levels (0.51 average). This validates that the model relies on retrieved biological priors rather than hallucinating or guessing.
- **Chain-of-Thought (CoT) Standardization:** Enabling CoT improves the mean score by approximately 15% (0.59 to 0.68). CoT acts as a regularizer, ensuring the model follows a logical inference path consistent with biological principles.

## 5 CONCLUSION

We demonstrate the efficacy of VCWorld in the task of predicting changes in gene expression following chemical perturbations. Our experiments show that VCWorld far outperforms current state-of-the-art black-box models. Its main contribution lies in aspects that are crucial for scientific utility. Through extensive ablation studies and case studies, we demonstrate that VCWorld is highly interpretable, with a human-readable, verifiable chain of reasoning for each prediction.

VCWorld still has room for improvement, with future directions including integrating a multi-agent framework to enhance autonomous reasoning and sample retrieval, extending generalization across diverse perturbation types such as gene, pathway, and combinatorial perturbations, and establishing more systematic benchmarks to comprehensively evaluate and compare specialized prediction models.

ACKNOWLEDGMENTS

This study has been supported by the National Natural Science Foundation of China 62041209, Natural Science Foundation of Shanghai 24ZR1440600, the Science and Technology Commission of Shanghai Municipality 24510714300, the Lingang Lab Fund LGL-8888.

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

## A  GENETAK

### A.1  DATA PROCESSING

We normalized all gene counts to $\log(TP10k+1)$ values (log-transformed UMI count per 10k), where the count $c_{ij}$ of gene $j$ in cell $i$ is mapped to:

$$\log\left(\frac{c_{ij}}{\sum_j c_{ij}} \cdot 10{,}000 + 1\right).$$

To focus on the most variably expressed genes, the top 2,000 highly variable genes (HVGs) were selected using the `Seurat v3` Python package, and retained in the expression matrix for downstream analyses, with the HVG list exported for reproducibility. To determine differentially expressed genes (DEGs), we ran the Wilcoxon signed-rank test (Wilcoxon, 1945) with Benjamini-Hochberg correction (Benjamini & Hochberg, 2000) between non-targeting control (NTC) cells and perturbed cells, for each perturbation.

We set $\text{FDR}_{\text{threshold}}$ and $\text{logFC}_{\text{threshold}}$ to screen perturbations with phenotypic effects, and $\text{PVAL}_{\text{threshold}}$ to screen negative samples (samples with no significant change in gene expression). Specifically, when

$$\text{FDR}_{\text{threshold}} \leq 0.05 \quad \& \quad \text{logFC}_{\text{threshold}} \geq 0.25$$

(i.e., more than 1.28-fold change in expression), we consider them as samples with phenotypic effects. Based on this, when

$$\text{PVAL}_{\text{neg\_threshold}} \geq 0.1,$$

we consider them as negative samples. In addition, we set $N_{\text{neg\_samples}} = 200$ to limit the number of negative samples, so as to avoid category imbalance.

### A.2  DISTRIBUTION OF GENETAK

To keep consistent with the experimental settings in the STATE paper, the dataset of each cell line was split into training and testing sets at a ratio of 30:70. During the experiment, VCWorld only retrieved samples from the training set. Further details regarding the dataset and data split statistics are provided in Table 4 and 5.

Table 4: Number of cells in each cell line.

| Cell line | Control | Perturbed | Sum |
|---|---|---|---|
| C32 | 29651 | 993514 | 1023165 |
| HepG2C3A | 20451 | 751387 | 771838 |
| HOP62 | 39683 | 1467690 | 1507373 |
| Hs 766T | 17802 | 695932 | 713734 |
| PANC-1 | 43878 | 1534738 | 1578616 |

Table 5: GeneTAK statistics for differentially expressed genes.

| Cell line | Split | Total genes | Non-DE genes | Differentially expressed genes | | |
|---|---|---|---|---|---|---|
| | | | | Total | Up-regulated | Down-regulated |
| C32 | Train | 128293 | 55200 | 73093 | 24746 | 48344 |
| | Test | 89961 | 38800 | 51161 | 16643 | 34518 |
| HepG2C3A | Train | 102253 | 55200 | 47053 | 16820 | 30233 |
| | Test | 72169 | 38800 | 33369 | 11629 | 21740 |
| HOP62 | Train | 154969 | 55200 | 99769 | 36615 | 63154 |
| | Test | 108777 | 38800 | 69977 | 25472 | 44505 |
| Hs 766T | Train | 101707 | 55200 | 46507 | 20774 | 25733 |
| | Test | 71891 | 38800 | 33091 | 14420 | 18671 |
| PANC-1 | Train | 163748 | 55200 | 108548 | 54081 | 54467 |
| | Test | 115568 | 38800 | 76768 | 38131 | 38637 |

# B    EXPERIMENT DETAILS

## B.1    METRICS

### B.1.1    ACCURACY

Accuracy measures the proportion of correctly predicted samples among all samples. It is defined as:

$$\text{Accuracy} = \frac{TP + TN}{TP + TN + FP + FN},$$

where $TP$, $TN$, $FP$, and $FN$ represent the numbers of true positives, true negatives, false positives, and false negatives, respectively.

**Definition of Confusion Matrix Components.**    Given binary ground-truth labels $y \in \{0, 1\}$ and model predictions $\hat{y} \in \{0, 1\}$:

$$TP = \sum_{i=1}^{N} \mathbb{I}(y_i = 1 \wedge \hat{y}_i = 1), \qquad TN = \sum_{i=1}^{N} \mathbb{I}(y_i = 0 \wedge \hat{y}_i = 0),$$

$$FP = \sum_{i=1}^{N} \mathbb{I}(y_i = 0 \wedge \hat{y}_i = 1), \qquad FN = \sum_{i=1}^{N} \mathbb{I}(y_i = 1 \wedge \hat{y}_i = 0),$$

where $\mathbb{I}(\cdot)$ is the indicator function.

### B.1.2    F1-SCORE

F1-score is the harmonic mean of Precision and Recall, balancing prediction correctness and coverage of true positives. The metrics are defined as:

$$\text{Precision} = \frac{TP}{TP + FP}, \qquad \text{Recall} = \frac{TP}{TP + FN},$$

$$\text{F1} = 2 \cdot \frac{\text{Precision} \cdot \text{Recall}}{\text{Precision} + \text{Recall}}.$$

### B.1.3  AUROC & AUPRC

The Area Under the Receiver Operating Characteristic Curve (AUROC) measures the model's ability to distinguish between classes. The True Positive Rate (TPR) and False Positive Rate (FPR) are defined as:

$$\text{TPR} = \frac{TP}{TP + FN}, \qquad \text{FPR} = \frac{FP}{FP + TN}.$$

AUROC is computed as:

$$\text{AUROC} = \int_0^1 \text{TPR}(FPR) \, d(FPR).$$

The Area Under the Precision–Recall Curve (AUPRC) quantifies the trade-off between Precision and Recall:

$$\text{AUPRC} = \int_0^1 \text{Precision}(Recall) \, d(Recall),$$

where Precision and Recall are computed as defined above.

### B.1.4  Q-SCORE

$$\text{Q-score} = 1 - \frac{AP}{QP + AP}$$

where $AP$ means the number of answered prompts, $QP$ means the number of prompts without a normal answer.

### B.2  BASELINES

All baselines were trained on 30% of the 45 cell lines from Tahoe-100m and 5 test cell lines, and all of our GeneTAK data was extracted from the remaining 70%, which eliminates the possibility of data leakage, Table5 is the result obtained from the inference of all baseline models on the five cell lines.

**GAT.** We moved beyond the disjoint subgraph design mentioned in the original manuscript. We integrated multiple knowledge sources (PubChem, DrugBank, GO, UniProt, Reactome) into a single connected graph structure, using standardized identifiers to align entities across databases. We used 70% of the labeled GeneTAK instances for training the GAT model, with the prediction task being a three-class classification comprising up, down, and no change.The source code of training is available at https://github.com/GENTEL-lab/VCWorld.

**CPA** is a framework to learn the effects of perturbations at the single-cell level. CPA encodes and learns phenotypic drug responses across different cell types, doses, and combinations. The source code of CPA is available at https://github.com/theislab/cpa. In our experiments, we use the official implementation with the hyperparameters.

**scVI** is a family of probabilistic models that perform many analysis tasks across single-cell, multi-omics, and spatial omics data. The source code of scVI is available at https://github.com/scverse/scvi-tools. In our experiments, we adopt the standard scVI model with the hyperparameters in the official GitHub repository.

**STATE** is a machine learning model that predicts cellular perturbation responses across diverse contexts. The source code of STATE is available at https://github.com/ArcInstitute/state. In our experiments, we use the official implementation with a LLaMA-style transformer backbone and the key hyperparameters.

### B.3  LARGE LANGUAGE MODELS

**Llama-3.1** is Meta's 8B-parameter instruction-tuned chat model optimized for multilingual dialogue, outperforming many open and closed models on standard benchmarks.The source code of Llama3.1-8B-Instruct is available at https://huggingface.co/meta-llama/Llama-3.1-8B-Instruct.

**Qwen2.5-7B-Instruct** is a 7.6B-parameter instruction-tuned open-source model with strong performance and efficiency on multilingual tasks, long-context understanding, and structured output

generation. The source code of Qwen2.5-7B-Instruct is available at `https://huggingface.co/Qwen/Qwen2.5-7B-Instruct`.

**Qwen3-4B-Instruct-2507** is the updated 4B-parameter Qwen3 instruction model with marked improvements in instruction following, reasoning, math, coding, and tool use, well-suited for lightweight multi-purpose deployment.The source code of Qwen3-4B-Instruct is available at `https://huggingface.co/Qwen/Qwen3-4B-Instruct-2507`.

**Qwen2.5-14B-Instruct** is a 14B-parameter instruction-tuned model in the Qwen2.5 family, offering significantly improved coding and math reasoning capabilities over Qwen2 for more demanding applications.The source code of Qwen2.5-14B-Instruct is available at `https://huggingface.co/Qwen/Qwen2.5-14B-Instruct`.

**Gemini-2.5-Flash** is Google's price-performance-optimized Gemini model with built-in "thinking" capabilities, designed for large-scale, low-latency, high-throughput and agentic workloads.

### B.4 INFERENCE PARAMETERS SETTINGS

For all models, we standardized the inference parameters with `temperature=0.6` and `top_k = 0.9` to balance creativity and determinism.

### B.5 COST AND LATENCY ANALYSIS

We monitored cost and latency using the OpenRouter API. To ensure a fair benchmark, the input prompt length was standardized to approximately 2,600 tokens for all models. The inference output tokens varied based on the reasoning length of each model. The detailed cost/latency breakdown for processing 1,000 samples is provided in Table 6.

Table 6: Model Cost and Latency(s)

| Model Name | Input Price | Output Price | Input Token | Output Token | Latency(s) | Total Cost($) |
|---|---|---|---|---|---|---|
| LLAMA3 8B | 0.03 | 0.06 | 2600 | 1300 | 0.31 | 0.156 |
| Qwen2.5-7B | 0.04 | 0.1 | 2600 | 2700 | 0.42 | 0.374 |
| Qwen2.5-14B | 0.05 | 0.22 | 2600 | 2700 | 0.51 | 0.724 |
| Qwen3-4B | 0 | 0 | 2600 | 2700 | 0.68 | 0 |
| Gemini-2.5-Flash | 0.30 | 2.50 | 2600 | 1400 | 0.56 | 4.28 |

Table 7: Statistics of predicted number of DEGs

| Task | Model | Count | C32 | HepG2C3A | HOP62 | Hs_766T | PANC-1 |
|---|---|---|---|---|---|---|---|
| | Truth | Number | 128293 | 102253 | 154969 | 101707 | 163748 |
| DE | STATE | Prediction | 157758 | 366238 | 317589 | 86684 | 316434 |
| | | Correct | 19488 | 33122 | 55717 | 9077 | 60640 |
| | CPA | Prediction | 21475 | 21204 | 19524 | 19838 | 19253 |
| | | Correct | 2424 | 2076 | 2755 | 1394 | 3178 |
| | scVI | Prediction | 460477 | 466868 | 449047 | 470606 | 435034 |
| | | Correct | 60973 | 40108 | 79858 | 38969 | 83801 |

### B.6 EVALUATION STRATEGY

Our evaluation strictly adheres to a zero-shot Drug Prediction protocol. Specifically:

- When predicting the effect of a test drug $p_{test}$, the retrieval corpus $\mathcal{D}$ consists exclusively of drugs from the training set.
- The model never has access to the ground truth or the entity of $p_{test}$ itself during retrieval.

- Inference relies solely on identifying functionally analogous cases (based on MoA and chemical structure) from the training set to extrapolate the potential effects of the unseen drug.

## B.7 KNOWLEDGE GRAPH-TO-TEXT PIPELINE FOR PROMPT GENERATION

To ensure the factual grounding and mitigate the risk of hallucination in our language model, we developed a deterministic pipeline to programmatically construct input prompts from multiple structured biological knowledge graphs (KGs). This process transforms machine-readable graph data into a human-readable, factual context that guides the model's summarization task. The pipeline consists of three primary stages: data integration and pre-processing, rule-based templated verbalization, and fact aggregation for prompt assembly.

### B.7.1 DATA INTEGRATION AND PRE-PROCESSING

Our pipeline integrates information from a diverse set of public biological knowledge bases, including Ensembl and UniProt for descriptive annotations, and graph-structured databases such as the Gene Ontology (GO), Reactome, CORUM, BioPlex, and STRING for relational information. Upon ingestion, a pre-processing step is applied to enhance data quality. Notably, we filter out high-degree nodes from relational graphs (*e.g.*, generic GO terms like "protein binding" or highly promiscuous interactors in STRING). This step reduces noise and focuses the downstream context on more specific and informative biological relationships.

### B.7.2 RETRIEVAL STRATEGY ANALYSIS

**Sensitivity Analysis.** To validate this choice, we conducted an ablation study increasing the neighborhood depth from 1 to 3 (Table 8). As shown in the additional results, increasing the depth significantly increased computational resource consumption and context length but resulted in negligible performance gains. This confirms that a 1-hop neighborhood is the optimal trade-off between efficiency and information density for this task.

Table 8: Accuracy Performance On Neighborhood Depth Variations Experiment

| Task | Hop | C32 | HepG2C3A | HOP62 |
|------|-----|------|----------|-------|
| **DE** | 1 | 0.70 | 0.68 | 0.71 |
| | 2 | 0.72 | 0.65 | 0.70 |
| | 3 | 0.69 | 0.67 | 0.70 |

### B.7.3 DECOUPLED RETRIEVAL STRATEGY

We employ a decoupled retrieval strategy for drugs and genes, utilizing a weighted combination of semantic and structural similarity.

**Drug Retrieval ($\alpha = 0.7$).** We assign a weight of $\alpha = 0.7$ to MoA-based semantic similarity and $1 - \alpha = 0.3$ to structure-based similarity (molecular fingerprints). This reflects the assumption that Mechanism of Action serves as the most direct reference for drug perturbation effects, while molecular structure provides complementary information related to scaffolds and functional groups.

**Gene Retrieval ($\alpha = 0.7$).** We set the weight for functional semantic annotation to $\alpha = 0.7$ and for PPI-based structural similarity to $1 - \alpha = 0.3$. Semantic annotations are prioritized due to their curated and generally high-confidence nature. Conversely, recent studies (Zhang et al., 2019) indicate that PPI networks may contain false-positive links. To mitigate noise, we assign a lower weight to the graph-derived component.

### B.7.4 RETRIEVAL-SET SIZE

We evaluate retrieval-set sizes of 5 and 10 (Table 9). The main effect lies in the model's *willingness to answer*. When the size is reduced to 5, the definitive response rate on the C32 cell line drops from

87% to 64%, as the model more frequently abstains due to insufficient evidence, effectively reducing hallucination but also degrading overall performance.

Table 9: Response Rate On Retrieval-set Size Experiment

| Task | Retrieval | C32 | HepG2C3A | HOP62 | Average |
|------|-----------|------|----------|-------|---------|
| **DE** | 5 | 0.64 | 0.67 | 0.60 | 0.64 |
|        | 10 | 0.87 | 0.87 | 0.88 | 0.87 |

## B.8 Rule-Based Templated Verbalization

The core of our methodology is a rule-based verbalization engine. Recognizing that each knowledge source possesses a unique data schema, we designed a specific template to translate its structured entries into natural language declarative statements.

For instance:

- A relationship from CORUM, representing `(Gene_A, is_member_of, Complex_X, in_cell_line_Y)`, is verbalized as: `Gene A is a member of the Complex X in cell line Y`.

- A protein-protein interaction from STRING, representing `(Gene_A, interacts_with, Gene_B)` with associated evidence types `[experimental, database]`, is translated to: `Based on evidence from experimental evidence and database evidence, Gene A may physically interact with Gene B`.

- Functional annotations from UniProt are extracted and formatted as statements like: `Functions of Gene A include: [function description]`.

This templated approach ensures that the conversion is consistent, deterministic, and preserves the semantic content of the original source.

## B.9 Fact Aggregation and Prompt Assembly

For each target gene, we aggregate declarative statements (facts) collected from all integrated sources into a single set. To automatically remove duplicates arising from overlapping databases, we use a set data structure. To keep the prompt concise and reduce the cognitive load on the model, we limit the maximum number of facts included, giving priority to high-confidence sources such as UniProt and curated pathway databases.

The resulting collection is then formatted as a bulleted list and inserted into a master prompt template under the heading "`Domain Knowledge:`". This final prompt conveys to the language model its role, a clearly defined task, operational constraints, and the essential, factually grounded context required to produce an accurate and reliable summary.

Given the imbalanced nature of the GeneTAK dataset, we agree that AUPRC provides a more informative assessment of performance on the minority class (differentially expressed genes).

## C Case Study

To qualitatively assess the explainability and reasoning capability of VCWorld, as well as its alignment with relevant wet-lab findings, we illustrate a representative case. The example focuses on the model's prediction of the effect of Larotrectinib on the gene MKI67.

**Larotrectinib** is a precision-targeted therapy that inhibits TRK kinase activity, thereby blocking the proliferative and survival signaling pathways driven by NTRK fusions in tumors. **MKI67** encodes the nuclear protein **Ki-67**, a well-established marker of cellular proliferation. High expression of Ki-67 indicates active cell division and is frequently used in oncology to assess tumor proliferation, guide treatment selection, and predict prognosis.

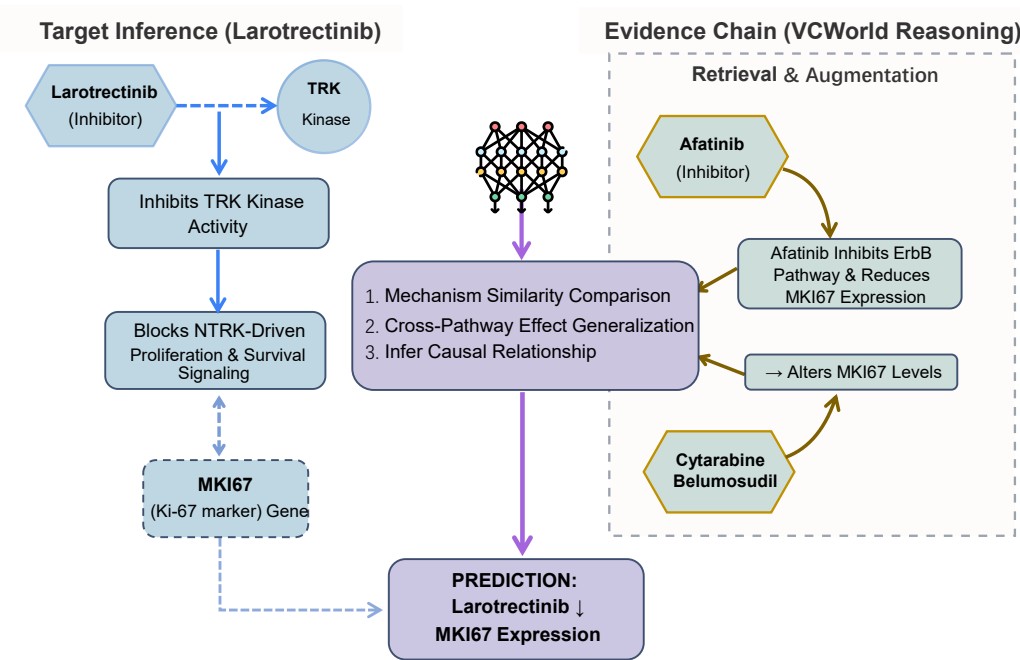

Figure 4: VCWorld Inference on (Larotrectinib,MKI67). The schematic illustrates the inference process for Larotrectinib-induced MKI67 downregulation, integrating direct target inference with retrieved biological evidence (BioContext). The LLM synthesizes these inputs via Chain-of-Thought (CoT) reasoning to predict causal relationships.

Although no direct evidence has previously linked **Larotrectinib** with **MKI67** expression, **VCWorld** provided a plausible reasoning pathway that supports their association (Figure 4). The model first retrieved Afatinib—a drug with a similar mechanism—as a positive reference. Evidence showing that Afatinib suppresses the ErbB signaling pathway and reduces MKI67 expression suggested that inhibition of upstream kinases can influence proliferative markers. Furthermore, perturbations induced by Cytarabine and Belumosudil also led to alterations in **MKI67** levels, reinforcing its role as a reliable indicator of anti-proliferative effects.

Based on this reasoning chain, VCWorld inferred that Larotrectinib downregulates MKI67 expression, a conclusion consistent with recent findings reporting that *"treatment with larotrectinib led to a reduction in proliferating cells (5.73% Ki67-positive cells)"* (Schmid et al., 2024). Consistently, immunofluorescence (IF) analysis revealed decreased Ki-67 signal intensity following Larotrectinib treatment (Kong et al., 2022).

Besides, we also show the original output results of the model used for reasoning about (Larotrectinib, MKI67), presented using the original markdown text.

```
# Answer for Drug: Larotrectinib, Gene: MKI67

1) **Which of the observed perturbing drugs are most similar to
↪  Larotrectinib (if any, including Larotrectinib itself)?**
The most similar observed perturbing drug to Larotrectinib is **Afatinib
↪  (Example 9)**.
*   **Larotrectinib:** Is an inhibitor of tropomyosin receptor kinases
↪  (Trk) A, B, and C. It prevents neurotrophin-Trk interaction and
↪  subsequent Trk activation, inhibiting downstream signaling pathways
↪  that regulate cell growth, differentiation, and survival.
*   **Afatinib:** Is n irreversible ErbB family blocker that covalently
↪  binds to and inhibits the kinase domains of EGFR, HER2, and HER4.
↪  This inhibits tyrosine kinase autophosphorylation, downregulating
↪  ErbB signaling, and affecting genes involved in cell proliferation
↪  and survival.
```

Both drugs are kinase inhibitors that target receptor tyrosine kinases
↪  (Trk in Larotrectinib's case, ErbB in Afatinib's case) and primarily
↪  impact downstream signaling pathways related to cell proliferation
↪  and survival.

2) **When using Larotrectinib or similar drugs, what downstream pathways
↪  or genes are differentially expressed? Justify your answer with the
↪  observed outcomes.**
Based on the description of Larotrectinib, it is expected to suppress the
↪  expression of genes involved in **cell growth, differentiation, and
↪  survival** that are regulated by Trk signaling.
The most similar drug, **Afatinib (Example 9)**, which also inhibits
↪  receptor tyrosine kinases involved in cell proliferation and
↪  survival, resulted in **differential expression of MKI67**. This
↪  indicates that perturbations of receptor tyrosine kinase pathways
↪  affecting cell growth and proliferation indeed lead to changes in
↪  MKI67 expression.
Other drugs impacting MKI67 (e.g., Belumosudil affecting STAT3/5,
↪  Cytarabine causing DNA damage/cell cycle control) also point to
↪  pathways that regulate cell proliferation and division as being
↪  differentially expressed.

3) **What perturbations of upstream pathways or genes result in
↪  differential expression of MKI67 or similar genes (if any)? Justify
↪  your answer with the observed outcomes.**
MKI67 is a marker of cell proliferation and is impacted by cell cycle
↪  perturbations. The examples show that various perturbations targeting
↪  cell growth, cell cycle control, and cellular stress responses result
↪  in differential expression of MKI67:
*   **Inhibition of growth-related receptor tyrosine kinases:**
↪  **Afatinib** (ErbB family blocker, affects cell proliferation and
↪  survival, Example 9) resulted in differential expression of MKI67.
*   **Modulation of Rho-associated coiled-coil containing protein kinases
↪  (ROCK) and STAT pathways:** **Belumosudil** (ROCK2 inhibitor,
↪  modulates STAT3/STAT5, Example 2) resulted in differential expression
↪  of MKI67. STAT pathways are often involved in cell growth and
↪  differentiation.
*   **Induction of DNA damage and replication stress:** **Cytarabine**
↪  (DNA polymerase inhibitor, causes DNA damage and replication stress,
↪  affecting cell cycle control, Example 10) resulted in differential
↪  expression of MKI67.
*   **Induction of oxidative stress/cellular stress response:**
↪  **Verteporfin** (photosensitizer, generates reactive oxygen species,
↪  inducing oxidative stress and cellular stress response, Example 6)
↪  resulted in differential expression of MKI67.
*   **Activation of metabolic pathways:** **Rosiglitazone** (PPAR-gamma
↪  agonist, affects glucose metabolism, Example 7) resulted in
↪  differential expression of MKI67, suggesting a link between
↪  metabolism and proliferation.
Perturbations that interfere with fundamental processes like DNA
↪  replication, cell cycle progression, growth signaling, or induce
↪  significant cellular stress tend to impact MKI67 expression.

4) Is a perturbation of Larotrectinib in MKI67 likely to result in
↪  differential expression of MKI67? For example, if 2) and 4) are
↪  unrelated or only indirectly related, it is unlikely we will observe
↪  differential expression. On the other hand, if 2) and 4)
↪  significantly overlap in specific genes or pathways, we may observe
↪  differential expression. Your final answer should end with one of
↪  these three options and nothing else.
Yes. Perturbation of Larotrectinib results in differential expression of
↪  MKI67.

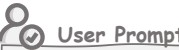 User Prompt

**In Hs766T cells, how does Tofacitinib affect the expression of FTH1 ?**

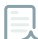 Generated Prompt

**[Start of Prompt]**
You are an expert molecular biologist who studies how small molecule drugs affect gene expression using Perturb-seq. Your goal is to determine: **Does a drug perturbation of Tofacitinib in Hs766T cells result in differential expression of FTH1?**
**You are given as input:**
Description of the drug (Tofacitinib): description of drug that is to perturb the cell
Description of gene of interest (FTH1): description of gene, the impact on which you wish to infer
Context: description of cell line in which the genes are expressed
Examples: set of experimental observations that describe the impact of small molecule perturbations on related genes, to contextualize your answer
**Output: Please extract the most relevant parts of the examples that address these five questions. Be specific.**
Which of the observed perturbing drugs are most similar to Tofacitinib (if any, including Tofacitinib itself)?
When using Tofacitinib or similar drugs, what downstream pathways or genes are differentially expressed? Justify your answer with the observed outcomes.
Which of the observed genes of interest are most similar to FTH1 (if any, including FTH1 itself)?
What perturbations of upstream pathways or genes result in differential expression of FTH1 or similar genes (if any)? Justify your answer with the observed outcomes.
Is a perturbation of Tofacitinib in FTH1 likely to result in differential expression of FTH1? For example, if 2) and 4) are unrelated or only indirectly related, it is unlikely we will observe differential expression. On the other hand, if 2) and 4) significantly overlap in specific genes or pathways, we may observe differential expression.
**Your final answer should end with one of these three options and nothing else.**
No. Perturbation of Tofacitinib does not impact FTH1.
Yes. Perturbation of Tofacitinib results in differential expression of FTH1.
There is insufficient evidence to determine how Perturbation of Tofacitinib affects FTH1.
**[End of Prompt]**

**[Start of Input]**
**Description of molecule drug (Tofacitinib):** Tofacitinib is a reversible Janus kinase (JAK) inhibitor that prevents the phosphorylation and activation of STAT transcription factors. This action blocks the JAK-STAT signaling pathway, which is involved in hematopoiesis and immune cell function. This perturbation may affect the transcription of cytokine-responsive genes that are regulated by the JAK-STAT pathway and are involved in inflammatory responses.
**Description of gene of interest (FTH1):** FTH1 is a ferritin heavy chain subunit enabling iron ion binding and sequestering, involved in iron storage and intracellular iron ion homeostasis. Iron overload, iron deficiency, or oxidative stress could impact FTH1 expression to regulate cellular iron levels.
**Context:** Hs 766T is a human pancreatic ductal adenocarcinoma (PDAC) cell line, established from the lymph node metastasis of a 73-year-old female patient. It carries hallmark pancreatic cancer alterations including KRAS G12D mutation, TP53 mutation, and CDKN2A inactivation, reflecting the canonical molecular landscape of PDAC. The cells grow adherently with an epithelial morphology, display a doubling time of ~60–70 hours, and are microsatellite stable (MSS). Hs 766T is widely used in studies of pancreatic tumor progression, metastasis, stromal interactions, and therapeutic resistance, and is included in large-scale resources such as the NCI-60 panel for pharmacogenomic profiling.
**Examples:**
**Example 1:**
Drug: c-Kit-IN-1
Gene: FTH1

# D    PROMPTS

We provide the prompts used by VCWorld and the model's response in the Appendix D. It clearly demonstrates that VCWorld is interpretable, capable of aligning with real-world biological knowledge, and has a traceable thinking process and final result.

Drug Description: c-Kit-IN-1 acts as an ATP-competitive inhibitor, binding to the kinase domain of the c-Kit receptor tyrosine kinase. This prevents the autophosphorylation of the receptor and the activation of downstream signaling pathways. Perturbation with c-Kit-IN-1 may affect the expression of genes regulated by the c-Kit signaling pathway, such as those in the MAPK and PI3K pathways.

Gene Description: FTH1 is a ferritin heavy chain subunit enabling iron ion binding and sequestering, involved in iron storage and intracellular iron ion homeostasis. Iron overload, iron deficiency, or oxidative stress could impact FTH1 expression to regulate cellular iron levels.

**Result: B) Perturbation of this drug results in differential expression of the gene of interest.**

**Example 2:**

Drug: AZD2858

Gene: FTH1

Drug Description: AZD2858 binds to the bromodomains of BET proteins, preventing them from binding to acetylated histones. This mechanism disrupts the transcriptional activation of key oncogenes, such as MYC. Perturbation with AZD2858 may affect the expression of BET target genes involved in cell proliferation.

Gene Description: FTH1 is a ferritin heavy chain subunit enabling iron ion binding and sequestering, involved in iron storage and intracellular iron ion homeostasis. Iron overload, iron deficiency, or oxidative stress could impact FTH1 expression to regulate cellular iron levels.

**Result: B) Perturbation of this drug results in differential expression of the gene of interest.**

**Example 3:**

Drug: BI-78D3

Gene: FTH1

Drug Description: BI-78D3 is a small molecule inhibitor of the protein-protein interaction between the transcription factor NRF2 and its negative regulator KEAP1. By disrupting this interaction, it modulates the activity of the NRF2 pathway. Perturbation with BI-78D3 may affect the expression of genes regulated by the NRF2 antioxidant response element.

Gene Description: FTH1 is a ferritin heavy chain subunit enabling iron ion binding and sequestering, involved in iron storage and intracellular iron ion homeostasis. Iron overload, iron deficiency, or oxidative stress could impact FTH1 expression to regulate cellular iron levels.

**Result: B) Perturbation of this drug results in differential expression of the gene of interest.**

**Example 4:**

Drug: Vemurafenib

Gene: FTH1

Drug Description: Vemurafenib is a competitive inhibitor of mutated BRAF-serine-threonine kinase, particularly the BRAF V600E mutation. It blocks the downstream MAPK signaling pathway to inhibit tumor growth and trigger apoptosis. This perturbation may affect the expression of genes regulated by the MAPK/ERK pathway, which are involved in cell growth and proliferation.

Gene Description: FTH1 is a ferritin heavy chain subunit enabling iron ion binding and sequestering, involved in iron storage and intracellular iron ion homeostasis. Iron overload, iron deficiency, or oxidative stress could impact FTH1 expression to regulate cellular iron levels.

**Result: A) Perturbation of this drug does not impact the gene of interest.**

**Example 5:**

Drug: Ciclopirox

Gene: FTH1

Drug Description: Ciclopirox is an antifungal agent thought to function by chelating polyvalent metal cations, which inhibits enzymes involved in processes like mitochondrial electron transport. This perturbation may affect the expression of genes involved in cellular metabolism, stress responses, DNA repair, and cell cycle regulation.

Gene Description: FTH1 is a ferritin heavy chain subunit enabling iron ion binding and sequestering, involved in iron storage and intracellular iron ion homeostasis. Iron overload, iron deficiency, or oxidative stress could impact FTH1 expression to regulate cellular iron levels.

**Result: B) Perturbation of this drug results in differential expression of the gene of interest.**

**Example 6:**

Drug: Afatinib

Gene: FTH1

Drug Description: Afatinib is an irreversible ErbB family blocker that covalently binds to the kinase domains of EGFR, HER2, and HER4. This binding irreversibly inhibits tyrosine kinase autophosphorylation, resulting in the downregulation of ErbB signaling. This perturbation may affect the expression of genes downstream of the ErbB signaling pathway that are involved in cell proliferation and survival.

Gene Description: FTH1 is a ferritin heavy chain subunit enabling iron ion binding and sequestering, involved in iron storage and intracellular iron ion homeostasis. Iron overload, iron deficiency, or oxidative stress could impact FTH1 expression to regulate cellular iron levels.

**Result: B) Perturbation of this drug results in differential expression of the gene of interest.**

**Example 7:**

Drug: BI-3406

Gene: FTH1

Drug Description: BI-3406 binds to the catalytic domain of SOS1, preventing it from engaging with and activating KRAS. This leads to the inhibition of KRAS signaling and a reduction in the proliferation of KRAS-dependent cancer cells. Perturbation with BI-3406 may affect the expression of genes downstream of the KRAS signaling pathway.

Gene Description: FTH1 is a ferritin heavy chain subunit enabling iron ion binding and sequestering, involved in iron storage and intracellular iron ion homeostasis. Iron overload, iron deficiency, or oxidative stress could impact FTH1 expression to regulate cellular iron levels.

**Result: B) Perturbation of this drug results in differential expression of the gene of interest.**

**Example 8:**

Drug: AZD1390

Gene: FTH1

Drug Description: AZD1390 is a potent inhibitor of Ataxia-Telangiectasia Mutated (ATM) kinase. This action prevents the activation of the DNA damage checkpoint and disrupts DNA repair processes. Perturbation with AZD1390 may affect the expression of genes involved in the DNA damage response and cell cycle checkpoints regulated by ATM kinase.

Gene Description: FTH1 is a ferritin heavy chain subunit enabling iron ion binding and sequestering, involved in iron storage and intracellular iron ion homeostasis. Iron overload, iron deficiency, or oxidative stress could impact FTH1 expression to regulate cellular iron levels.

**Result: A) Perturbation of this drug does not impact the gene of interest.**

**Example 9:**

Drug: Palmatine (chloride)

Gene: FTH1

Drug Description: Palmatine (chloride) has multiple mechanisms of action, including acting as a poison for topoisomerases, which leads to DNA damage. It also inhibits indoleamine 2,3-dioxygenase 1 (IDO-1), an enzyme involved in immune tolerance. Perturbation with Palmatine (chloride) may affect the expression of genes involved in DNA replication and repair, as well as immune response pathways.

Gene Description: FTH1 is a ferritin heavy chain subunit enabling iron ion binding and sequestering, involved in iron storage and intracellular iron ion homeostasis. Iron overload, iron deficiency, or oxidative stress could impact FTH1 expression to regulate cellular iron levels.

**Result: A) Perturbation of this drug does not impact the gene of interest.**

**Example 10:**

Drug: Regorafenib

Gene: FTH1

Drug Description: Regorafenib is a small molecule inhibitor of multiple membrane-bound and intracellular kinases, including VEGFR, KIT, PDGFR, FGFR, and RAF. This broad inhibition targets pathways involved in oncogenesis, tumor angiogenesis, and the maintenance of the tumor microenvironment. This perturbation may affect the expression of genes regulated by these various kinase signaling pathways that are involved in cell proliferation and angiogenesis.

Gene Description: FTH1 is a ferritin heavy chain subunit enabling iron ion binding and sequestering, involved in iron storage and intracellular iron ion homeostasis. Iron overload, iron deficiency, or oxidative stress could impact FTH1 expression to regulate cellular iron levels.

**Result: A) Perturbation of this drug does not impact the gene of interest.**

[End of Input]

[Start of Output]

1)

2)

3)

4)

5)

[End of Output]

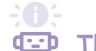 **Thoughts**

**Investigating Tofacitinib's Impact**
I'm focusing on Tofacitinib, a JAK inhibitor, and its potential to influence FTH1 expression in Hs766T cells. I've begun by examining Tofacitinib's mechanism, specifically its role in blocking JAK-STAT signaling pathways. This understanding will be crucial in predicting its effect on FTH1, the target gene. **My next step involves researching the link between JAK-STAT and iron metabolism to understand the downstream effects on FTH1 expression.**

**Exploring Drug Similarities**
I'm now comparing Tofacitinib to the example drugs, focusing on their mechanisms of action. I see c-Kit-IN-1, a tyrosine kinase inhibitor, as potentially relevant due to its kinase-inhibiting nature, though not directly targeting JAK-STAT. The other examples seem quite different, acting on epigenetic regulators or cellular stress responses. I'll need to assess if shared downstream pathways might indirectly link them to FTH1 regulation.

**Comparing Drug Kinase Inhibitors**
I'm deep into the analysis of kinase inhibitors. Several example drugs share this mechanism with Tofacitinib, though they target distinct kinase families. I've noted c-Kit-IN-1, Vemurafenib, Afatinib, AZD1390 and Regorafenib as particularly relevant, due to their inhibition of kinases involved in various signal transduction pathways. I'm focusing now on identifying potential shared downstream pathways that might connect these drugs to FTH1 regulation, despite their diverse primary targets.

**Refining Kinase Connections**
Now I'm delving deeper into the potential links between the kinase inhibitors and FTH1. Given the examples, I see that while Tofacitinib directly blocks JAK-STAT, other drugs like c-Kit-IN-1, Vemurafenib, Afatinib, AZD1390, and Regorafenib might indirectly influence FTH1 through shared downstream pathways. I'm focusing on those pathways activated or inhibited by these kinase drugs to evaluate their relevance to FTH1 regulation. This will necessitate cross-referencing published literature.

**Identifying JAK-STAT Targets**
I'm now focusing on identifying the specific cytokine-responsive genes regulated by the JAK-STAT pathway, which is affected by Tofacitinib. The goal is to determine if any of these target genes might indirectly influence FTH1 expression. While no drugs are directly analogous, some kinase inhibitors (c-Kit-IN-1, Afatinib) showed FTH1 differential expression, highlighting the need to understand potential shared pathways. I'm cross-referencing literature to link Tofacitinib's pathway to FTH1's regulators.

**Evaluating Pathway Links**
I'm now focusing on how the JAK-STAT pathway, targeted by Tofacitinib, might intersect with FTH1 regulation. Considering the lack of direct JAK-STAT inhibitors in the examples, **I'm examining if inflammatory responses, a key feature of JAK-STAT signaling, could indirectly influence iron homeostasis and FTH1 expression. This will require further literature exploration.**

**Linking JAK-STAT to Iron**
I'm now focusing on how the JAK-STAT pathway, which is targeted by Tofacitinib, might intersect with iron regulation in Hs766T cells. While I've examined cytokine-responsive genes, I'm now exploring potential connections between inflammatory responses and FTH1 expression, requiring extensive literature review.

**Evaluating JAK-STAT Impact**
The connection between Tofacitinib's JAK-STAT pathway and FTH1 regulation is under review. I'm focusing on the inflammatory responses driven by this pathway, as these might influence iron homeostasis. I'm also examining the literature regarding STAT transcription factor regulation of iron-related genes to predict FTH1 expression shifts.

**Predicting Tofacitinib's Effect**
I'm now integrating information on JAK-STAT and iron regulation to predict if Tofacitinib will alter FTH1 in Hs766T. **Given that the JAK-STAT pathway influences inflammatory responses and iron metabolism, it appears plausible that FTH1 expression will shift with Tofacitinib.** This requires further review.

I've determined that Tofacitinib, a JAK-STAT inhibitor, has a plausible link to FTH1 regulation in the Hs766T cells, primarily through the effects of inflammation on iron homeostasis. This requires further scrutiny, as there is currently no direct evidence from the examples.

**Analyzing JAK-STAT's Influence**
I'm now considering how Tofacitinib, which inhibits the JAK-STAT pathway, interacts with iron regulation. Although there are no direct JAK-STAT inhibitors among the examples, this is a plausible mechanism. FTH1 is known to be regulated by oxidative stress, metal chelation, and the KRAS pathway, all of which are linked.

**The link between Tofacitinib, a JAK-STAT inhibitor, and FTH1 in Hs766T cells is clear. The JAK-STAT pathway is involved in inflammation, which impacts iron homeostasis. KRAS, deregulated in this cell line, feeds into the JAK-STAT pathway, which is shown to impact FTH1.**

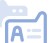 **Answer**

**Which of the observed perturbing drugs are most similar to Tofacitinib (if any, including Tofacitinib itself)?**

Tofacitinib is a Janus kinase (JAK) inhibitor that blocks the JAK-STAT signaling pathway. None of the drugs in the provided examples are JAK inhibitors. The most similar class of drugs are other kinase inhibitors, although they target different pathways. These include:
c-Kit-IN-1: A c-Kit receptor tyrosine kinase inhibitor.
Vemurafenib: A BRAF serine-threonine kinase inhibitor.
Afatinib: An ErbB family tyrosine kinase inhibitor.
AZD1390: An ATM kinase inhibitor.
Regorafenib: A multi-kinase inhibitor targeting VEGFR, KIT, PDGFR, FGFR, and RAF.

**When using Tofacitinib or similar drugs, what downstream pathways or genes are differentially expressed? Justify your answer with the observed outcomes.**

Tofacitinib is described as affecting cytokine-responsive genes regulated by the JAK-STAT pathway. The provided examples do not include drugs that target this pathway, so direct evidence is unavailable. The outcomes for other kinase inhibitors are inconsistent:
Perturbation with c-Kit-IN-1 (affecting MAPK and PI3K pathways) and Afatinib (affecting ErbB signaling) resulted in differential expression of FTH1.
Perturbation with Vemurafenib (affecting MAPK/ERK), AZD1390 (affecting DNA damage response), and Regorafenib (affecting multiple pathways including RAF) did not result in differential expression of FTH1.
This indicates that inhibition of some, but not all, kinase signaling pathways can lead to differential expression of FTH1.

**Which of the observed genes of interest are most similar to FTH1 (if any, including FTH1 itself)?**
The only observed gene of interest in all provided examples is FTH1 itself.

**What perturbations of upstream pathways or genes result in differential expression of FTH1 or similar genes (if any)? Justify your answer with the observed outcomes.**

Based on the examples, differential expression of FTH1 results from perturbations of several distinct pathways:
Kinase Signaling: Inhibition of c-Kit (Example 1: c-Kit-IN-1) and ErbB family kinases (Example 6: Afatinib).
KRAS Signaling: Inhibition of the SOS1-KRAS interaction (Example 7: BI-3406). This is particularly relevant as the Hs766T cell line has a KRAS mutation.
Antioxidant/Stress Response: Modulation of the NRF2 pathway (Example 3: BI-78D3), which directly regulates genes involved in oxidative stress, including FTH1.
Metabolic Stress/Ion Homeostasis: Chelation of metal cations (Example 5: Ciclopirox), which directly relates to FTH1's function in iron storage.
Epigenetic Regulation: Inhibition of BET bromodomain proteins (Example 2: AZD2858), which affects broad transcriptional programs.

**Is a perturbation of Tofacitinib in FTH1 likely to result in differential expression of FTH1?** For example, if 2) and 4) are unrelated or only indirectly related, it is unlikely we will observe differential expression. On the other hand, if 2) and 4) significantly overlap in specific genes or pathways, we may observe differential expression.

The pathways impacted by Tofacitinib (JAK-STAT, inflammation, cytokine signaling) are highly related to pathways shown to regulate FTH1 (stress response, KRAS signaling). There is a well-established biological link between inflammation, cytokine signaling (mediated by JAK-STAT), and the regulation of iron homeostasis, where FTH1 is a key player. Furthermore, in the context of KRAS-mutated pancreatic cancer cells like Hs766T, the oncogenic KRAS pathway is known to drive inflammatory cytokine production, creating a dependency on JAK-STAT signaling. Since the examples show that inhibiting KRAS signaling directly results in differential expression of FTH1 (Example 7), it is highly probable that inhibiting the downstream JAK-STAT pathway with Tofacitinib will also result in differential expression of FTH1.

**Yes. Perturbation of Tofacitinib results in differential expression of FTH1.**

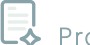 Prompt

You are an expert molecular biologist who studies how drug affect the cellular reaction using Perturb-seq.

Task: You are writing a brief overview of the small molecule drug{{pert}}, with a focus on its MoA and Targets. You will be provided a set of database entries about the drug. Ensure that your overview remains faithful to this domain knowledge.

Format:
- Write one to two sentences describing the primary MoA and target of {{pert}}.
- Write one sentence describing the potential downstream impact of perturbing drug {{pert}}.

Constraints:
- Maintain a professional tone throughout.
- Do not comment on your own writing.
- Do not add any notes or references. Do not make up additional information.
- Do not discuss the importance or impact of the gene. Focus only on its function.

Domain knowledge:
{{entries}}

Brief overview of drug {{pert}}:

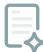 Prompt

You are an expert molecular biologist who studies how genes are related using Perturb-seq.

Task: You are writing a brief overview of the human gene {{name}}, with a focus on its molecular and cellular functions. You will be provided with a set of database entries about the gene. Ensure that your overview remains faithful to this domain knowledge.

Format:
- Write one to two sentences describing the primary molecular and cellular function of gene {{name}}.
- Write one sentence describing what types of perturbations might impact the expression of gene {{name}}. For example, you might consider pathways that are upstream of the gene or compensatory mechanisms.

Constraints:
- Maintain a professional tone throughout.
- Do not comment on your own writing.
- Do not add any notes or references. Do not make up additional information.
- Do not discuss the importance or impact of the gene. Focus only on its function.

Domain knowledge:
{{entries}}

Brief overview of gene {{name}}:

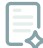 Prompt

You are an expert molecular biologist who studies how drug affect the cellular reactions using Perturb-seq.

Task: You are writing a brief overview of the small molecule drug{{pert}}, with a focus on the downstream effects of perturbing {{name}} .

Inputs: You are provided
- Description of perturbing drug {{pert}}
- Database entries relating {{pert}} to other drugs or targets.

Format:
- Write up to five sentences describing the molecular and cellular impact of perturbing drug{{pert}}.

Constraints:

- Omit the importance or impact of the gene. Focus only on its function.
- Omit all non-specific information and obvious statements, e.g. "this gene is involved in cellular processes."
- Remain faithful to all domain knowledge. Do not make up additional information.
- Maintain a professional tone throughout. Do not comment on your own writing. Do not add any notes or references.

Description of drug {{pert}}: {{summary}}

Relations to other drugs/targets:
{{entries}}

Downstream effects of perturbing drug {{pert}} :

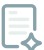 Prompt

You are an expert molecular biologist who studies how genes are related using Perturb-seq.

Task: You are writing a brief overview of the human gene {{name}}, with a focus on molecular and cellular perturbations that may affect the levels of gene {{name}}. For example, you might consider pathways that are upstream of the gene or compensatory mechanisms.

Inputs: You are provided
- Description of gene of interest {{name}}
- Database entries relating {{name}} to other genes or pathways

Format:
- Write up to five sentences describing potential molecular and cellular perturbations that may impact the levels of {{name}}.

Constraints:
- Remain faithful to all domain knowledge. Do not make up additional information.
- Summarize all common aspects succinctly, but point out notable differences within these sets of genes.
- Maintain a professional tone throughout. Do not comment on your own writing. Do not add any notes or references.
- Omit the importance or impact of the gene. Focus only on its function.
- Omit all non-specific information and obvious statements, e.g. "this gene is involved in cellular processes."

Description of gene {{name}}: {{summary}}

Relations to other genes:
{{entries}}

Perturbations that may affect the levels of {{name}}:

