# OpenReview forum: "VCWorld: A Biological World Model for Virtual Cell Simulation"
_ICLR.cc/2026/Conference — ICLR 2026 Poster_

### Official Review · Reviewer_rJsH · 2025-10-25

**Soundness:** 2
**Presentation:** 2
**Contribution:** 2
**Rating:** 2
**Confidence:** 3

**Summary:**

VCWorld is a framework for simulating cellular perturbation responses, integrating a structured biological knowledge graph with the reasoning of Large Language Models (LLMs). Using a Chain-of-Thought (CoT) process, it generates step-by-step, interpretable predictions. The paper also introduces GeneTAK, a benchmark curated from the Tahoe-100M dataset, and claims state-of-the-art, interpretable performance.

**Strengths:**

The design, which combines a knowledge graph with LLM reasoning, seems a novel approach that improves LLM performance on perturbation prediction.

**Weaknesses:**

- STATE has a particularly low performance, which contradicts the reported performance in [1]. The reason remains unclear and can arise from suboptimal implementation or insufficient training data. This should be better clarified in the work.
- The contribution is unclear. GeneTAK should not be emphasized as a contribution since it is just reorganizing an existing database. Current claims exaggerate the contribution of the work.
- It is unclear how the results compare to well-established benchmarks in [1-2].
- Importantly, additional results with modern reasoning language models should be presented, such as the results with more widely used LLMs. From the current results, the real usefulness of VCWorld remains unclear; maybe simply using the latest generation of LLMs will remove the need of the knowledge base.

[1] Adduri, Abhinav K., et al. "Predicting cellular responses to perturbation across diverse contexts with State." bioRxiv(2025): 2025-06.
[2] Ahlmann-Eltze, Constantin, Wolfgang Huber, and Simon Anders. "Deep-learning-based gene perturbation effect prediction does not yet outperform simple linear baselines." Nature Methods (2025): 1-5.

**Questions:**

See weaknesses.

---

> ### Author Response · Authors · 2025-11-22
> **Thank the reviewer for the valuable questions and suggestions. Answer to Weakness 1 & 2**
>
> Thank you for your thoughtful and constructive review of our paper. We greatly appreciate your feedback and are pleased to know that you find the work's strength in the idea and illustrations. Upon review, it seems there may have been a misunderstanding, which could have contributed to the lower score. We provide the following response to address your concern.
>
> ## Weakness 1 - Clarification on STATE Performance
>
> **Q:** STATE shows particularly low performance compared to the results reported in [1]. The discrepancy is unclear and may arise from suboptimal implementation or insufficient training data.
>
> **A:**
> We appreciate the reviewer's attention to this point. Since STATE is a primary baseline for VCWorld, we conducted a carefully controlled evaluation. Our experimental setup matches the few-shot configuration used in STATE exactly, including the same five GeneTAK cell lines and the 30/70 data split. We employed the officially released Tahoe weights from [HuggingFace]([https://huggingface.co/arcinstitute/ST-Tahoe]
> ) and repeated all experiments under three random seeds. The obtained performance closely matches that reported in the original paper, confirming correct implementation.
>
> To explain the weaker performance of STATE on GeneTAK, we conducted the following analysis:
>
> 1. Task formulation differences.
>    STATE models cell-level transcriptomic responses, whereas our setting focuses on gene-specific expression directionality prediction. These are inherently distinct tasks.
>
> 2. Differences in evaluation metrics.
>    STATE evaluates performance using correlation metrics (Pearson/Spearman/PR) at the cell level. In contrast, classification metrics such as Accuracy are known to be challenging for all foundation models, consistent with our results. After adding AUC-ROC and AUPRC in our updated experiments, the comparison with STATE becomes more balanced.
>
> 3. Differences in HVG selection strategies.
>    We applied Seurat v3 highly variable gene filtering separately for each cell line. The STATE paper does not disclose its HVG selection procedure, which likely contributes to performance discrepancies. Our filtering code is:
>
>    ```python
>    adata_dis_num = sc.pp.highly_variable_genes(
>        adata, flavor="seurat v3",
>        n_top_genes=2000,
>        subset=False, inplace=False
>    )
>     ```
>
> ## Weakness 2 - Contribution of GeneTAK
>
> **Q:** The contribution is unclear. GeneTAK should not be emphasized as a contribution since it is just reorganizing an existing database. Current claims exaggerate the contribution of the work.
>
> **A:**
> We sincerely appreciate this important comment. GeneTAK is not a simple reorganization of existing data; rather, it represents a methodological transformation tailored to the VCWorld task. Its contribution is reflected in the following aspects:
>
> 1. Transforming a sparse regression problem into a gene-centric classification task.
>    The Tahoe-100M matrix exhibits extreme sparsity and extensive dropout noise, making it difficult for conventional metrics to distinguish true biological signals. GeneTAK reformulates the problem into gene-level differential expression classification, yielding more reliable and interpretable evaluation outcomes.
>
> 2. Statistically rigorous data derivation rather than mere reformatting.
>    GeneTAK is constructed through Wilcoxon signed-rank testing with Benjamini–Hochberg correction, combined with stringent dual thresholds for statistical significance (FDR ≤ 0.05) and effect size (|logFC| ≥ 0.25). Additionally, a PVAL ≥ 0.1 criterion is applied to ensure balanced negative samples. These procedures guarantee that labels capture genuine biological perturbation effects.
>
> 3. Enabling fine-grained and biologically meaningful evaluation.
>    Standard baselines (e.g., scVI, STATE) tend to predict 3-5× more DEGs than ground truth, a phenomenon often hidden by conventional benchmarks. By emphasizing gene-level Precision and Recall, GeneTAK exposes such over-prediction artifacts, providing a more stringent testing ground for evaluating generalization capability.
>
> In summary, GeneTAK supplies the first dedicated dataset for assessing drug-induced gene-level perturbation effects. It offers a statistically validated, biologically grounded, and fine-grained foundation that is essential for the VCWorld framework and for advancing perturbation modeling research.

---

> > ### Author Response · Authors · 2025-11-22
> > **Answer to  Weakness 3**
> >
> > ## Weakness 3 - Comparison with Benchmarks [1-2]
> >
> > **Q:** It is unclear how the results compare to well-established benchmarks in [1-2].
> >
> > **A:**
> > Thank you for highlighting this important point. We provide detailed comparisons with both STATE [1] and Linear Baselines [2], addressing differences in model formulation, evaluation procedures, and experimental outcomes.
> >
> > ### (1) Comparison with STATE (Adduri et al., 2025)
> >
> > STATE outputs continuous gene-expression values, whereas GeneTAK defines binary differential expression labels. To enable a fair comparison, we transformed STATE's predictions into binary DE/non-DE labels using the same Wilcoxon signed-rank test procedure employed during GeneTAK construction. Our findings (Supplementary Table 5) show that STATE predicts substantially more DEGs than ground truth  (e.g., 366k vs. 33k for HepG2C3A),  which results in low Precision (0.13) and F1-score (0.17).
> >
> > The VCWorld framework, by incorporating biological context and reasoning mechanisms, effectively reduces over-prediction and achieves **Precision 0.59 and F1-score 0.63**.
> >
> > ### (2) Comparison with Linear Baselines (Ahlmann-Eltze et al., 2025)
> >
> > Ahlmann-Eltze et al. emphasize that deep learning models frequently fail to outperform simple linear baselines in perturbation prediction tasks. Our results reinforce this observation: deep models such as scVI and CPA do not significantly surpass the RANDOM baseline (0.50) on the Directional Change (DIR) task. This underscores the challenges inherent in regression-based perturbation prediction and motivates our classification-based GeneTAK formulation.
> >
> > Under this more robust formulation, VCWorld consistently outperforms simple baselines while addressing the core limitations discussed in [1] and [2]. Our analysis thus provides both quantitative comparison and conceptual clarification regarding benchmark differences.

---

> > > ### Author Response · Authors · 2025-11-22
> > > **Answer to Weakness 4**
> > >
> > > ## Weakness 4 - Value of VCWorld vs. Modern LLMs
> > >
> > > **Q:** Additional results with modern reasoning language models should be presented. From the current results, the real usefulness of VCWorld remains unclear; using the latest generation of LLMs might remove the need for the knowledge base.
> > >
> > > **A:**
> > > We thank the reviewer for this impactful question. To address whether stronger LLMs diminish the need for an explicit knowledge base, we performed extensive ablation studies and added additional models (Table 1-2). Our findings demonstrate that both reasoning and biological context are indispensable.
> > >
> > > 1. Scaling experiments with modern Qwen models. We incorporated multiple Qwen-2.5 models known for strong reasoning capabilities. A clear performance gradient emerges as model size increases, and the Qwen-2.5-14B model achieves performance comparable to the Gemini-2.5-Flash model reported in the main text. This validates both reproducibility and the ability of VCWorld to effectively leverage stronger base models.
> > >
> > > 2. Knowledge-free ablations reveal significant performance degradation.Both the VCWorld-no-BioContext variant and the Qwen-no-BWK variant exhibit notable performance drops. This confirms that, despite exposure to biological knowledge during pretraining, LLMs still require explicit, context-specific prompting linked to the current cell line and drug. Thus, stronger LLMs do not eliminate the necessity of the BioContext knowledge module.
> > >
> > > 3. Retrieval-Only baseline demonstrates the necessity of reasoning.
> > >    We added a Retrieval-Only baseline that performs nearest-neighbor voting without LLM reasoning. The full VCWorld model significantly outperforms this baseline, indicating that retrieval alone is insufficient. Effective perturbation prediction requires the integration of retrieved knowledge through reasoning—a central component of VCWorld.
> > >
> > > Together, these results show that the utility of VCWorld arises from the synergy between retrieval, biological context encoding, and LLM-based reasoning. Model strength alone cannot replace this combination, and the knowledge base remains essential for achieving accurate and biologically grounded predictions.
> > >
> > > Table 1 : Accuracy Performence On DE Task
> > > | Task | Model | C32 | HepG2C3A | HOP62 | Hs_766T | PANC-1 |
> > > | :--- | :--- | :--- | :--- | :--- | :--- | :--- |
> > > | **DE** | Random | 0.50 | 0.50 | 0.50 | 0.50 | 0.50 |
> > > | | GAT | 0.62 | 0.54 | 0.64 | 0.57 | 0.58 |
> > > | | CPA | 0.27 | 0.18 | 0.21 | 0.30 | 0.17 |
> > > | | ScVI | 0.66 | 0.48 | 0.64 | 0.61 | 0.68 |
> > > | | State | 0.165 | 0.38 | 0.41 | 0.09 | 0.47 |
> > > | | **VCWorld** (Retrieval) | 0.57 | 0.53 | 0.58 | 0.57 | 0.58 |
> > > | | **VCWorld** (w/o Biocontext) | 0.52 | 0.54 | 0.51 | 0.52 | 0.50 |
> > > | | **VCWorld** (w/o CoT) | 0.61 | 0.57 | 0.59 | 0.60 | 0.56 |
> > > | | **VCWorld** (Llama3-8B) | 0.35 | 0.36 | 0.41 | 0.36 | 0.39 |
> > > | | **VCWorld** (Qwen3-4B-Instruct) | 0.41 | 0.47 | 0.46 | 0.49 | 0.46 |
> > > | | **VCWorld** (Qwen2.5-7B-Instruct) | 0.54 | 0.54 | 0.55 | 0.58 | 0.64 |
> > > | | **VCWorld** (Qwen2.5-14B-Instruct)| 0.65 | 0.66 | **0.72** | 0.67 | **0.76** |
> > > | | **VCWorld** (Gemini-2.5-flash) | **0.70** | **0.68** | 0.71 | **0.68** | 0.61 |
> > >
> > > Table 2 : Accuracy Performence On DIR Task
> > > | Task | Model | C32 | HepG2C3A | HOP62 | Hs_766T | PANC-1 |
> > > | :--- | :--- | :--- | :--- | :--- | :--- | :--- |
> > > | **DIR** | Random | 0.5 | 0.5 | 0.5 | 0.5 | 0.5 |
> > > | | GAT | 0.58 | 0.61 | 0.54 | 0.55 | 0.52 |
> > > | | CPA | 0.22 | 0.19 | 0.17 | 0.21 | 0.15 |
> > > | | ScVI | 0.47 | 0.46 | 0.41 | 0.40 | 0.38 |
> > > | | State | 0.49 | 0.51 | 0.50 | 0.55 | 0.51 |
> > > | | **VCWorld** (Retrieval) | 0.47 | 0.47 | 0.47 | 0.50 | 0.51 |
> > > | | **VCWorld** (w/o Biocontext) | 0.37 | 0.36 | 0.36 | 0.17 | 0.34 |
> > > | | **VCWorld** (w/o CoT) | 0.42 | 0.45 | 0.51 | 0.47 | 0.45 |
> > > | | **VCWorld** (Llama3-8B) | 0.37 | 0.34 | 0.38 | 0.40 | 0.35 |
> > > | | **VCWorld** (Qwen3-4B-Instruct) | 0.54 | 0.55 | 0.52 | 0.48 | 0.50 |
> > > | | **VCWorld** (Qwen2.5-7B-Instruct) | 0.56 | 0.56 | 0.55 | 0.51 | 0.53 |
> > > | | **VCWorld** (Qwen2.5-14B-Instruct)| 0.67 | 0.59 | 0.65 | **0.71** | 0.66 |
> > > | | **VCWorld** (Gemini-2.5-flash) | **0.72** | **0.68** | **0.67** | 0.66 | **0.69** |
> > >
> > >
> > > ## Reference
> > > [1] Adduri, Abhinav K., et al. "Predicting cellular responses to perturbation across diverse contexts with State." bioRxiv(2025): 2025-06.
> > >
> > > [2] Ahlmann-Eltze, Constantin, Wolfgang Huber, and Simon Anders. "Deep-learning-based gene perturbation effect prediction does not yet outperform simple linear baselines." Nature Methods (2025): 1-5.

---

### Official Review · Reviewer_inaw · 2025-10-29

**Soundness:** 2
**Presentation:** 3
**Contribution:** 2
**Rating:** 4
**Confidence:** 4

**Summary:**

This paper introduces VCWorld, a system for predicting gene expression changes following drug perturbations in cells. The authors reformulate the traditional cell-level regression problem into a gene-centric binary classification framework addressing two tasks: differential expression detection (DE) and directional change prediction (DIR). VCWorld integrates a biological knowledge graph built from seven databases with LLM-based retrieval and chain-of-thought (CoT) reasoning using Gemini 2.5 Flash to produce mechanistic explanations. A new benchmark, GeneTAK, derived from Tahoe-100M, includes five cell lines and 348 drug perturbations with a 30:70 train-test split by perturbation. VCWorld achieves an average accuracy of 0.68 on both tasks, outperforming existing deep learning baselines: STATE (0.30–0.51), scVI (0.42–0.61), and CPA (0.23–0.44).

The experimental methodology shows both strengths and weaknesses. Positively, the work uses appropriate statistical testing (Wilcoxon signed-rank), clear dataset construction, extensive ablations, and transparent reporting. However, several issues limit validity. The term “world model” is used incorrectly; in established RL literature (Ha & Schmidhuber 2018; Hafner et al. 2020), world models learn temporal dynamics, which VCWorld does not. Key baselines are missing: no graph neural networks (GCN, GAT, GraphSAGE) on the same knowledge graph, no classical ML using graph features, no simpler retrieval pipeline without LLM reasoning. Consequently, it is unclear whether performance gains stem from graph structure, LLM reasoning, or both. While the ablations are informative, the large performance gap between Gemini 2.5 Flash and Llama3-8B (0.68 to 0.37 DE accuracy) without cost or compute analysis raises reproducibility concerns. Finally, the “white-box” interpretability claim is unvalidated; the paper includes no quantitative audit verifying that reasoning traces correspond to real biological mechanisms.

The paper is well-organized and clearly written, with strong visual presentation (pipeline diagram and tables). The GeneTAK dataset and labeling procedures are described in adequate detail, and the appendices provide useful prompt examples. Nonetheless, the “world model” framing is misleading and should be replaced with a more accurate term such as knowledge-grounded retrieval-augmented classification system. The novelty is somewhat overstated: VCWorld is essentially retrieval-augmented generation (RAG) applied to biological perturbation prediction, combining known components (knowledge graph integration, hybrid retrieval, CoT reasoning) rather than introducing a new algorithm. Discussion of existing RAG literature could be expanded, and the paper should include clearer descriptions of knowledge-graph maintenance, conflict resolution, and update policies.

Empirically, the paper contributes valuable resources and results; methodologically, it is modest. Strengths include the GeneTAK benchmark, the gene-centric reformulation improving data efficiency, strong empirical performance demonstrating that knowledge-grounded reasoning outperforms purely data-driven models, and interpretable natural-language reasoning outputs. However, the architecture primarily reuses established components: graph construction, hybrid semantic-structural retrieval, contrastive case retrieval, and CoT prompting (Wei et al. 2022). There are no fundamentally new algorithms or theoretical insights. The system’s dependence on proprietary Gemini 2.5 Flash, with much weaker results for Llama3-8B (0.37 DE, 0.56 DIR) limits reproducibility and accessibility. Without graph-based or classical ML baselines, it remains unclear whether LLM reasoning is necessary or whether the knowledge graph alone drives performance improvements.

**Strengths:**

1.	Addresses an important problem in computational biology with strong empirical performance: VCWorld reaches 0.68 accuracy on both DE and DIR, while existing models perform near chance on DIR.

2.	GeneTAK provides a useful community benchmark with rigorous construction and consistent evaluation protocols.

3.	Gene-centric formulation effectively mitigates data sparsity and improves interpretability.

4.	Provides mechanistic textual explanations alongside predictions, improving transparency over black-box deep models.

5.	Thorough ablation studies demonstrate component importance: removing BioContext drops accuracy to ~0.51, removing CoT to ~0.59, and replacing Gemini 2.5 with Llama3-8B to ~0.37 (DE).

6.	Integrates structured biological knowledge from seven authoritative databases in a coherent, data-efficient system.

**Weaknesses:**

1.	VCWorld does not learn temporal dynamics or simulate cellular trajectories and this misframing creates false expectations about the system's capabilities. Please revise throughout.

2.	Missing critical baselines: no GNNs, traditional ML with graph features, or retrieval-only ablations.

3.	Overstated methodological novelty: essentially standard RAG adapted to a biological context.

4.	Interpretability claims unvalidated: no systematic check that explanations align with known biology.

5.	Missing analyses: computational cost, statistical significance, error analysis by perturbation type, and retrieval-parameter sensitivity.

6.	Unusual 30:70 train-test split is only lightly justified.

7.	Heavy dependence on Gemini 2.5 Flash without open alternatives demonstrated limits reproducibility and accessibility for most researchers.

**Questions:**

1.	Can you provide GNN baselines (GCN, GAT, GraphSAGE) using your knowledge graph?
2.	How does performance compare when using smaller fine-tuned models?
3.	Have you validated mechanistic reasoning against literature, what fraction of explanations are biologically correct?
4.	What are the computational and API costs per query relative to baselines?
5.	Can you ablate the LLM entirely (graph-based k-NN or scoring) to test the necessity of language reasoning?
6.	How does accuracy vary with retrieval-set size or when retrieving only positive/negative examples?
7.	Provide failure analysis by perturbation and gene coverage; how are out-of-graph entities handled?
8.	Replace the “world model” terminology with a more precise descriptor. Major issue which needs to be addressed for acceptance.
This paper offers solid empirical advances, especially the GeneTAK benchmark and strong results showing knowledge-grounded reasoning can outperform data-hungry deep learning. However, several weaknesses must be addressed: misleading terminology, incomplete baselines (no GNNs or retrieval-only models), overstated novelty, lack of cost analysis, and unvalidated interpretability claims. With revisions adding graph-based and classical baselines, testing open LLMs, validating reasoning quality, reporting costs, and reframing the contribution as a knowledge-grounded retrieval-augmented classifier, the work would merit acceptance.
I will consider raising it to 5 if authors revise the misleading world model framing and perform GNN baseline tests. I would further raise it to 6 (accept) if authors perform cost analysis and address interpretability claims.

**Details Of Ethics Concerns:**

The work uses public single-cell datasets and purely computational methods with no human or animal subjects; no ethical issues are apparent.

---

> ### Author Response · Authors · 2025-11-22
> **Thank the reviewer for the valuable questions and suggestions. Answer to Weakness 1**
>
> Thank you for your thoughtful and constructive review of our paper. We greatly appreciate your feedback and are pleased to know that you find the work's strength in quality, clarity and significance.We provide the following response to address your concern.
> ## Weakness 1
> Q: VCWorld does not learn temporal dynamics or simulate cellular trajectories and this misframing creates false expectations about the system's capabilities.
>
> A: We thank the reviewer for the rigorous distinction between "Simulation" and "Dynamics." We acknowledge that VCWorld does not model continuous temporal trajectories (e.g., via ODEs). However, we wish to clarify that our terminology aligns with prevalent definitions in the deep generative modeling literature, and we emphasize that VCWorld provides a **"logical simulation."**
>
> 1.  Alignment with Perturbation Modeling Standards: In existing high-throughput perturbation literature (e.g., scGen [1], CPA [2]), the term "Simulation" is standardly used to describe the prediction of **State Transitions**-specifically, the direct mapping from a control state to a perturbed state ($State_{control} \rightarrow State_{perturbed}$)-rather than continuous time-series dynamics. VCWorld adopts this established paradigm to ensure comparability with existing baselines.
> 2.  Simulation of Mechanistic Steps: Although VCWorld does not simulate explicit time steps, it uniquely simulates the mechanistic steps of biological responses. Through Chain-of-Thought (CoT) reasoning, VCWorld explicitly models the causal chain: $\text{Drug} \rightarrow \text{Target} \rightarrow \text{Pathway} \rightarrow \text{Gene Response}$. This distinguishes VCWorld from "black-box" generative models that perform opaque, direct mappings.
> 3.  Data Constraints: It is important to note that current large-scale datasets (e.g., Tahoe-100M) consist primarily of endpoint measurement snapshots rather than time-series data. Learning ground-truth continuous trajectories from such static data is infeasible for any current model. We strongly agree with the reviewer that integrating temporal dynamics is a critical future direction, but it depends on the availability of high-temporal-resolution Perturb-seq datasets.
>
> To avoid ambiguity, we will explicitly clarify in the manuscript that our "simulation" refers to state transition modeling and causal reasoning, not temporal dynamics.

---

> > ### Author Response · Authors · 2025-11-22
> > **Answer to Weakness 2 & Question 1 & 5**
> >
> > ## Weakness 2 & Question 1 & 5
> > Q: Missing critical baselines: no GNNs, traditional ML with graph features, or retrieval-only ablations. [Can you provide GNN baselines (GCN, GAT, GraphSAGE) using your knowledge graph?] [Can you ablate the LLM entirely (graph-based k-NN or scoring) to test the necessity of language reasoning?]
> >
> > We sincerely appreciate the constructive suggestion to include GNN and retrieval-only baselines. To rigorously address this, we constructed a **Unified Heterogeneous Graph** to implement Graph Attention Networks (GAT) and conducted pure retrieval ablation studies. These additions significantly expand our evaluation (see Table 1 and Table 2).
> >
> > 1.  GNN Baselines on a Unified Graph: To support GNN training, we moved beyond the disjoint subgraph design mentioned in the original manuscript. We integrated multiple knowledge sources (PubChem, DrugBank, GO, UniProt, Reactome) into a single connected graph structure, using standardized identifiers to align entities across databases. This explicitly connects drugs, targets, pathways, and genes, providing the topological foundation for GNNs.
> >     *   Result Analysis: As shown in the new results, the GAT baseline demonstrates strong performance, significantly outperforming the generative baseline STATE. This confirms that the graph topology contains valuable signals. However, GAT still underperforms compared to VCWorld. This reveals a key limitation of pure graph models: they struggle to bridge the modality gap and connection sparsity between the chemical (drug) and biological (gene) spaces. While GNNs rely on message passing along existing edges, VCWorld's LLM leverages semantic reasoning and open-world knowledge to bridge missing links, proving that "topology alone is not enough."
> >
> > 2.  Retrieval-Only Ablation: To test if the LLM merely parrots retrieved results, we implemented a retrieval-only baseline using a k-NN voting system based on our hybrid similarity metric.
> >     *   Result Analysis: VCWorld significantly outperforms the retrieval-only baseline. This demonstrates that simply retrieving similar historical experiments is insufficient. The model requires reasoning capabilities to synthesize evidence, validating VCWorld's core value proposition.
> >
> > **Table 1: Accuracy Performance on DE (Differential Expression) Task**
> > | Task | Model | C32 | HepG2C3A | HOP62 | Hs_766T | PANC-1 |
> > | :--- | :--- | :--- | :--- | :--- | :--- | :--- |
> > | **DE** | Random | 0.50 | 0.50 | 0.50 | 0.50 | 0.50 |
> > | | GAT | $0.62 \pm .03$ | $0.54 \pm .02$ | $0.64 \pm .04$ | $0.57 \pm .01$ | $0.58 \pm .01$ |
> > | | CPA | 0.27 | 0.18 | 0.21 | 0.30 | 0.17 |
> > | | ScVI | 0.66 | 0.48 | 0.64 | 0.61 | 0.68 |
> > | | State | $0.17 \pm .02$ | $0.38 \pm .01$ | $0.41 \pm .01$ | $0.09 \pm .02$ | $0.47 \pm .01$ |
> > | | **VCWorld** (Retrieval-only) | $0.57 \pm .01$ | $0.53 \pm .00$ | $0.58 \pm .01$ | $0.57 \pm .02$ | $0.58 \pm .01$ |
> > | | **VCWorld** (w/o Biocontext) | 0.52 | 0.54 | 0.51 | 0.52 | 0.50 |
> > | | **VCWorld** (w/o CoT) | 0.61 | 0.57 | 0.59 | 0.60 | 0.56 |
> > | | **VCWorld** (Llama3-8B) | $0.35 \pm .01$ | $0.36 \pm .00$ | $0.41 \pm .01$ | $0.36 \pm .00$ | $0.39 \pm .02$ |
> > | | **VCWorld** (Qwen3-4B-Instruct) | $0.41 \pm .00$ | $0.47 \pm .01$ | $0.46 \pm .00$ | $0.49 \pm .02$ | $0.46 \pm .01$ |
> > | | **VCWorld** (Qwen2.5-7B-Instruct) | $0.54 \pm .01$ | $0.54 \pm .00$ | $0.55 \pm .00$ | $0.58 \pm .01$ | $0.64 \pm .02$ |
> > | | **VCWorld** (Qwen2.5-14B-Instruct)| 0.65 | 0.66 | **0.72** | 0.67 | **0.76** |
> > | | **VCWorld** (Gemini-2.5-flash) | **0.70** | **0.68** | 0.71 | **0.68** | 0.61 |
> >
> > **Table 2: Accuracy Performance on DIR (Directionality) Task**
> > | Task | Model | C32 | HepG2C3A | HOP62 | Hs_766T | PANC-1 |
> > | :--- | :--- | :--- | :--- | :--- | :--- | :--- |
> > | **DIR** | Random | 0.5 | 0.5 | 0.5 | 0.5 | 0.5 |
> > | | GAT | $0.58 \pm .03$ | $0.61 \pm .01$ | $0.54 \pm .02$| $0.55 \pm .01$ | $0.52 \pm .01$ |
> > | | CPA | 0.22 | 0.19 | 0.17 | 0.21 | 0.15 |
> > | | ScVI | 0.47 | 0.46 | 0.41 | 0.40 | 0.38 |
> > | | State | $0.49 \pm .01$ | $0.51 \pm .03$ | $0.50 \pm .01$ | $0.55 \pm .02$ | $0.51 \pm .01$ |
> > | | **VCWorld** (Retrieval-only) | $0.47 \pm .01$ | $0.47 \pm .02$ | $0.47 \pm .01$ | $0.50 \pm .00$ | $0.51 \pm .01$ |
> > | | **VCWorld** (w/o Biocontext) | 0.37 | 0.36 | 0.36 | 0.17 | 0.34 |
> > | | **VCWorld** (w/o CoT) | 0.42 | 0.45 | 0.51 | 0.47 | 0.45 |
> > | | **VCWorld** (Llama3-8B) | $0.37 \pm .01$ | $0.34 \pm .02$ | $0.38 \pm .04$ | $0.40 \pm .01$ | $0.35 \pm .010$ |
> > | | **VCWorld** (Qwen3-4B-Instruct) | $0.54 \pm .01$ | $0.55 \pm .00$ | $0.52 \pm .00$ | $0.48 \pm .02$ | $0.50 \pm .01$ |
> > | | **VCWorld** (Qwen2.5-7B-Instruct) | $0.56 \pm .02$ | $0.56 \pm .01$ | $0.55 \pm .00$ | $0.51 \pm .00$ | $0.53 \pm .01$ |
> > | | **VCWorld** (Qwen2.5-14B-Instruct)| 0.67 | 0.59 | 0.65 | **0.71** | 0.66 |
> > | | **VCWorld** (Gemini-2.5-flash) | **0.72** | **0.68** | **0.67** | 0.66 | **0.69** |

---

> > > ### Author Response · Authors · 2025-11-22
> > > **Answer to Weakness 3&4  & Question 3**
> > >
> > > ## Weakness 3
> > > Q: Overstated methodological novelty: essentially standard RAG adapted to a biological context.
> > >
> > > A: We thank the reviewer for questioning the methodological novelty. While VCWorld indeed employs a "Retrieve-then-Generate" architecture similar to standard RAG, we contend that it represents a task-specific reconstruction rather than a simple adaptation. Standard RAG is designed for open-domain QA using static text, whereas VCWorld is engineered for causal analogical reasoning in sparse biological spaces. The differences are structural and critical:
> > >
> > > 1.  Dynamic Ground Truth vs. Static Knowledge: Standard RAG typically retrieves static knowledge snippets. In contrast, VCWorld retrieves dynamic experimental results (Ground Truth labels from historical experiments), enabling the model to capture cell-type-specific biological responses that static knowledge graphs cannot provide.
> > > 2.  Dual-Set Retrieval vs. Top-K: Unlike the simple Top-K strategy of standard RAG, VCWorld introduces a Dual-Set Retrieval Mechanism. We explicitly retrieve two disjoint support sets: the Positive Set ($S_{analog}$, effective perturbations) and the Negative Set ($S_{contrast}$, ineffective perturbations). This functions as a structured Few-shot prompting strategy designed to reconstruct the decision boundary for the LLM. By simultaneously presenting "what caused a change" and "what did not," we provide necessary logical constraints for the binary classification task, which standard RAG cannot achieve.
> > > 3.  Graph-Guided Hybrid Similarity: Standard RAG relies on dense vector semantic similarity. VCWorld implements a graph-guided hybrid similarity metric, necessitated by the heterogeneity of biological data. We leverage the topological structure of the knowledge graph (e.g., pathway distance, chemical structure) to identify mechanistically relevant samples even when semantic text descriptions are sparse or missing. This explicit injection of topological priors into retrieval ranking significantly differs from standard semantic RAG.
> > >
> > > ## Weakness 4 & Question 3
> > > Q: Interpretability claims unvalidated: no systematic check that explanations align with known biology. [Have you validated mechanistic reasoning against literature, what fraction of explanations are biologically correct?]
> > >
> > > A: We thank the reviewer for this insight regarding interpretability. Through case studies across various domains (see Table 3), we found that VCWorld provides **Scientific Verifiability**, which offers higher practical value to biologists. VCWorld reveals traceable logical paths strictly based on retrieved evidence, allowing researchers to verify not just the prediction, but the premises of the reasoning.
> > >
> > > To qualitatively assess this, we present a specific case study: predicting the effect of **Larotrectinib** (a highly selective TRK inhibitor) on **MKI67** (a cell proliferation marker). (The full Chain-of-Thought output is available via [https://drive.google.com/file/d/1BfhJXRdj67-eb55K6nC3FtRRXFfFGN6G](https://drive.google.com/file/d/1BfhJXRdj67-eb55K6nC3FtRRXFfFGN6G)).
> > >
> > > 1.  Evidence-Based Reasoning: VCWorld's reasoning is grounded in retrieved similar biological contexts. In this case, the model retrieved **Afatinib** (an ErbB inhibitor) as a positive example from the training set. Although targeting different kinases, the CoT correctly identified their shared downstream effects. The model inferred that Larotrectinib inhibits the Trk pathway, analogous to Afatinib's inhibition of the ErbB pathway—both key drivers of cell proliferation. Consequently, it reasoned that blocking these upstream kinases attenuates proliferation signals, leading to the downregulation of MKI67. It further cited other retrieved perturbations (e.g., Cytarabine) to confirm MKI67 as a reliable indicator of anti-proliferative effects.
> > > 2.  Alignment with Clinical Facts: The model concluded that Larotrectinib downregulates MKI67 expression. This is biologically accurate and supported by clinical data. Clinical trials [] demonstrate that Larotrectinib achieves high remission rates in TRK fusion-positive tumors, with pathology confirming a significant decrease in the Ki-67 index (the protein encoded by MKI67) post-treatment.
> > >
> > > This case demonstrates that the generated explanations align causally with established biological principles: "Kinase Inhibition $\rightarrow$ Blocked Proliferation $\rightarrow$ Marker Downregulation." This transparent, logic-based approach allows biologists to treat model outputs as scientifically reasoned hypotheses rather than mere statistical probabilities.
> > >
> > > **Table 3: Selected Case Studies for Interpretability**
> > > | Area | Case |
> > > | :--- | :--- |
> > > | Chemotherapy Stress Response | (Gemcitabine, FN1) [3] |
> > > | Targeted Therapy Signaling | (Pemigatinib, EGR1) [4] |
> > > | Feedback Loops | (Everolimus, FGFR2) [5] |

---

> > > > ### Author Response · Authors · 2025-11-22
> > > > **Answer to Weakness 5 &6 &7 & Question 4 & 2**
> > > >
> > > > ## Weakness 5 & Question 4
> > > > **Q: Missing analyses: computational cost, statistical significance, error analysis by perturbation type, and retrieval-parameter sensitivity. [What are the computational and API costs per query relative to baselines?]**
> > > >
> > > > **(1) Computational Cost Analysis**
> > > > We monitored costs using OpenRouter.ai. While the input size is fixed (2600 tokens), output tokens vary. The costs and latency for 1000 queries are detailed below. Note that open-source models (e.g., Qwen) can be deployed locally for zero API cost.
> > > >
> > > > **Table 4: Model Cost and Latency (per 1000 queries)**
> > > > *Note: Input/Output Prices are generally per 1M tokens.*
> > > > | Model Name | Input Price ($) | Output Price ($) | Input Token | Output Token | Latency(s) | Total Cost ($) |
> > > > | :--- | :--- | :--- | :--- | :--- | :--- | :--- |
> > > > | LLAMA3 8B | 0.03 | 0.06 | 2600 | 1300 | 0.31 | 0.156 |
> > > > | Qwen2.5-7B | 0.04 | 0.1 | 2600 | 2700 | 0.42 | 0.374 |
> > > > | Qwen2.5-14B | 0.05 | 0.22 | 2600 | 2700 | 0.51 | 0.724 |
> > > > | Qwen3-4B | 0 | 0 | 2600 | 2700 | 0.68 | 0  |
> > > > | Gemini-2.5-flash | 0.30 | 1.50 | 2600 | 1400 | 0.56 | 4.28 |
> > > >
> > > > **(2) Statistical Significance**
> > > > We are completing inference runs with three random seeds (see Tables 1 & 2). Current results show that VCWorld exhibits excellent robustness on GeneTAK. Furthermore, the consistent improvement across all 5 cell lines (C32, HepG2C3A, HOP62, Hs 766T, PANC-1) serves as intuitive evidence of robustness across varying biological contexts. Full statistical details will be included in the final manuscript.
> > > >
> > > > **(3) Error Analysis by Perturbation Type**
> > > > Our case studies reveal a positive correlation between VCWorld's performance and the richness of the **Biocontext**.
> > > > *   **High Performance:** For well-documented drugs like Busulfan, Resveratrol, and Capmatinib, accuracy in DE tasks is high (0.80, 0.75, 0.72, respectively).
> > > > *   **Lower Performance:** For drugs with missing MoA or target information (e.g., Thymopentin, TAK-733, Penfluridol), accuracy drops (0.25, 0.35, 0.42).
> > > > This confirms the critical importance of Biocontext availability for the reasoning process.
> > > >
> > > > **(4) Retrieval-Parameter Sensitivity**
> > > > We separate retrieval for drugs and genes with distinct parameter settings:
> > > > *   **Drug Retrieval:** We set the MoA biological mechanism weight $\alpha=0.7$ and the structural (fingerprint) weight $(1-\alpha)=0.3$. This prioritizes detailed biological mechanisms for reasoning while using chemical structure to provide heuristic insights related to functional groups.
> > > > *   **Gene Retrieval:** We set the functional semantic annotation weight $\alpha=0.7$ and the PPI network structural weight $(1-\alpha)=0.3$. We prioritize semantic annotations as they are manually curated and high-confidence, whereas PPI networks are known to contain significant false-positive connections [6], prompting us to lower the weight of graph-based similarity.
> > > >
> > > > ## Weakness 6
> > > > **Q: Unusual 30:70 train-test split is only lightly justified.**
> > > >
> > > > This split strategy was chosen to accommodate generative baseline models (e.g., STATE) which require a portion of the data for few-shot training/context. For VCWorld, this "training" set serves as the database for retrieval. We maintained this split to ensure a fair comparison where all models have access to the same amount of prior information.
> > > >
> > > > ## Weakness 7 & Question 2
> > > > **Q: Heavy dependence on Gemini 2.5 Flash without open alternatives demonstrated limits reproducibility and accessibility for most researchers. [How does performance compare when using smaller fine-tuned models?]**
> > > >
> > > > We appreciate this concern regarding accessibility. We have supplemented our results with the **Qwen series** of open-source models (see Table 1, Table 2), which are renowned for their reasoning capabilities.
> > > > *   **Performance Gradient:** Different parameter sizes show a clear performance gradient.
> > > > *   **Reproducibility:** Notably, the **Qwen-2.5-14B** version demonstrates performance comparable to the Gemini 2.5-flash model used in the main text.
> > > > This verifies the reproducibility of our results and proves that the VCWorld framework effectively leverages advanced open-source foundation models, mitigating reliance on proprietary APIs.

---

> > > > > ### Author Response · Authors · 2025-11-22
> > > > > **Answer to Question 6 & 7 & 8**
> > > > >
> > > > > ## Question 6
> > > > > **Q: How does accuracy vary with retrieval-set size or when retrieving only positive/negative examples?**
> > > > >
> > > > > **(1) Retrieval-Set Size:**
> > > > > We tested sizes of 5 and 10. The primary impact is on the model's **willingness to answer**. With a size of 5, the rate of definitive answers on the C32 cell line dropped from 87% to 64%, as the model tends to judge the evidence as insufficient (abstention) to prevent hallucination. Consequently, overall performance metrics naturally decline.
> > > > >
> > > > > **Table 5: Response Rate vs. Retrieval-Set Size**
> > > > > | Task | Retrieval Size | C32 | HepG2C3A | HOP62 | Average |
> > > > > | :--- | :--- | :--- | :--- | :--- | :--- |
> > > > > | **DE** | 5 | 0.64 | 0.67 | 0.60 | 0.64 |
> > > > > | | 10 | 0.87 | 0.87 | 0.88 | 0.87 |
> > > > >
> > > > > **(2) Positive-Only vs. Balanced Retrieval:**
> > > > > We tested balanced (Positive/Negative) vs. Positive-only retrieval. Results indicate that retrieving only positive samples introduces a label bias that outweighs the experimental context, leading the LLM to make aggressive predictions. The model becomes overly prone to predicting differential expression, causing accuracy to degrade toward random guessing (see Table 6).
> > > > >
> > > > > **Table 6: Performance with Positive-Only vs. Balanced Samples**
> > > > > | Task | Sample Type | C32 | HepG2C3A | HOP62 | Average |
> > > > > | :--- | :--- | :--- | :--- | :--- | :--- |
> > > > > | **DE** | Positive Only | 0.62 | 0.50 | 0.58 | 0.57 |
> > > > > | | Positive & Negative | 0.87 | 0.87 | 0.88 | 0.87 |
> > > > >
> > > > > ## Question 7
> > > > > **Q: Provide failure analysis by perturbation and gene coverage; how are out-of-graph entities handled?**
> > > > >
> > > > > **Failure Analysis:**
> > > > > *   **Node Degree Correlation:** We acknowledge that performance correlates with the node degree (richness of known information) of entities in the KG.
> > > > > *   **Abandonment Rate (Q-Score):** For entities with sparse knowledge, the LLM tends to **refuse to answer** rather than hallucinate. We view this high abandonment rate in knowledge-poor scenarios as a protective mechanism ensuring reliability.
> > > > >
> > > > > **Out-of-Graph (OOG) Entities:**
> > > > > *   **Coverage:** Our constructed KG integrates vast biological knowledge, covering the majority of entities in standard datasets. OOG instances are rare.
> > > > > *   **Handling:** When OOG entities occur, we rely on the LLM's internal knowledge to infer general features (e.g., using SMILES strings to infer drug similarity or sequence/family information to infer gene similarity), allowing the model to function even without explicit graph nodes.
> > > > >
> > > > > ## Question 8
> > > > > **Q: Replace the "world model" terminology with a more precise descriptor.**
> > > > >
> > > > > We thank the reviewer for the feedback regarding terminology precision. We understand that "World Model" is traditionally associated with learning temporal dynamics in RL or Computer Vision. However, we used this term to align with emerging literature in *AI for Science*. For instance, Istrate et al. (2025) explicitly propose "Biological World Models" to describe systems using biological priors as validators for reasoning. In VCWorld, the "world" is instantiated not by video frames, but by the causal rules of the cellular environment simulated via the Knowledge Graph and LLM engine.
> > > > >
> > > > > To avoid ambiguity, we will adopt a more precise descriptor in the Method section, such as **"Knowledge-Augmented Cellular Simulator,"** rather than the broader "World Model." We will also add a clarification in the supplementary materials defining our usage as a simulation of the static causal environment rather than time-series dynamics.
> > > > >
> > > > >
> > > > > ## References
> > > > > [1] Lotfollahi, Mohammad, F. Alexander Wolf, and Fabian J. Theis. "scGen predicts single-cell perturbation responses." Nature methods 16.8 (2019): 715-721.
> > > > >
> > > > > [2] Lotfollahi, Mohammad, et al. "Predicting cellular responses to complex perturbations in high-throughput screens." Molecular systems biology 19.6 (2023): e11517.
> > > > >
> > > > > [3] Binenbaum, Y., Na'ara, S., & Gil, Z. (2015). Gemcitabine resistance in pancreatic ductal adenocarcinoma. *Drug Resistance Updates*, 23, 55-68.
> > > > >
> > > > > [4] Chesnokov, M. S., Yadav, A., & Chefetz, I. (2022). Optimized transcriptional signature for evaluation of MEK/ERK pathway baseline activity and long-term modulations in ovarian cancer. *International Journal of Molecular Sciences*, 23(21), 13365.
> > > > >
> > > > > [5] Szymczyk, Jakub, et al. "FGF/FGFR-dependent molecular mechanisms underlying anti-cancer drug resistance." Cancers 13.22 (2021): 5796.
> > > > >
> > > > > [6] Zhang, Zhaopeng, et al. "Predicting essential proteins from protein-protein interactions using order statistics." Journal of Theoretical Biology 480 (2019): 274-283.

---

### Official Review · Reviewer_Vroo · 2025-11-01

**Soundness:** 3
**Presentation:** 2
**Contribution:** 2
**Rating:** 6
**Confidence:** 3

**Summary:**

This paper introduces VCWorld, a novel computational framework for predicting cellular responses to drug perturbations. First, VCWorld reframes the prediction task, moving from a high-dimensional regression problem (predicting exact gene levels) to a gene-level binary classification task. To generate a prediction, the model retrieves relevant information from the constructed KG, along with relevant experiments from its training data, and then feeds the information to the LLM, which generates a step-by-step CoT to support its final answer.

**Strengths:**

1. The idea of generating a CoT rationale to provides a human-readable, verifiable hypothesis for the model prediction is original;
2. The paper introduces GeneTAK, a novel benchmark that simplifies the prediction proble;
3. By leveraging a pre-existing KG, it can reason about new drugs or genes as long as they are present in its knowledge base.

**Weaknesses:**

1. The gene-centric, one-at-a-time prediction model raises scalability concerns. This formulation also ignores the cell's pre-perturbation state and treats genes as independent entities, which is not biologically accurate.
2. The KG's construction is described at a high level. It is not explained if the resulting graph consists of a single connected network or many disconnected pieces, which impacts the "graph-based structural similarity" metric.
3. The model works by retrieving similar past experiments to guide its reasoning. It is not explained why this was this chosen over a more direct Graph-RAG approach that would retrieve causal paths directly from the KG to explain the (drug, gene) relationship.
4. Finally, the paper presents its generated reasoning (the CoT) as evidence of interpretability. However, a generated explanation is not a guaranteed, faithful transcript of the model's actual internal process. Therefore, **reasoning and interpretability should not be treated as interchangeable**.

**Questions:**

1. I think state does not use only 5 cell lines in tahoe, but it is trained instead on entire 50. Please clarify the setup to ensure fairness of comparison.

2.The model uses a hybrid similarity (part text, part structure) to find similar experiments. Why is this complex metric necessary, and what is its impact compared to a simpler, purely semantic retrieval?

---

> ### Author Response · Authors · 2025-11-22
> **Thank the reviewer for the valuable questions and suggestions. Answer to Weakness 1 & 2**
>
> Thank you for your thoughtful and constructive review of our paper. We provide the following response to address your concern.
> ## Response to Weakness 1
> Q: The gene-centric, one-at-a-time prediction model raises scalability concerns. This formulation also ignores the cell's pre-perturbation state and treats genes as independent entities, which is not biologically accurate.
>
> A:
> We thank the reviewer for this insightful comment. We address your concerns regarding scalability and the biological formulation from two perspectives:
>
> (1) Scalability of the Model
> Although our model adopts a gene-centric prediction approach, we demonstrate that scalability is effectively guaranteed through parallel computing and optimized task formulation.
> *   Parallelization: The independence of gene prediction tasks allows us to leverage large-scale parallel computing efficiently. Unlike models requiring long-sequence processing, our queries can be distributed across massive GPU clusters or handled via high-concurrency APIs. Our stress tests in an environment with 8 NVIDIA A100 GPUs confirm that the model remains stable under high-concurrency requests.
> *   Cost vs. Benefit: While the computational cost of LLM inference is higher than simple regression baselines, we argue this is a necessary trade-off to achieve White-box Interpretability and Chain-of-Thought (CoT) reasoning.
> *   Practicality: In real-world drug screening scenarios, researchers typically focus on High-Variable Genes (HVGs, such as the top 2,000 used in our paper) or specific pathway genes. This reduces the runtime to a range that is highly practical for both academic research and industrial applications.
>
> (2) Clarification on Pre-perturbation State
> We wish to clarify that while the *output* is generated gene-by-gene, the *inference process* does not treat genes as isolated entities, nor does it ignore the cell state.
> *   Semantic Network Simulation: When constructing the inference prompt, we retrieve the neighborhood subgraph of the target gene from the Knowledge Graph (KG), including upstream regulators and protein-protein interaction partners. Therefore, when predicting Gene A, the model implicitly incorporates its interactions with Genes B and C. This simulates the gene regulatory network at a semantic level.
> *   Semantic State: We do not ignore the "pre-perturbation state"; rather, we upgrade it from a "numerical vector" to a "Semantic Description." As shown in Appendix C, we input the cell line's background knowledge (e.g., KRAS mutation status in Hs 766T, tissue origin) as context. Compared to single-cell sequencing vectors, which are often plagued by noise and batch effects, this Semantic State provides a more robust reasoning baseline for the LLM, ensuring that predictions remain cell-type specific.
> ## Response to Weakness 2
> Q: The KG's construction is described at a high level. It is not explained if the resulting graph consists of a single connected network or many disconnected pieces, which impacts the "graph-based structural similarity" metric.
>
> A:
> We appreciate the reviewer's question regarding the connectivity of the Knowledge Graph (KG). This is indeed crucial for understanding our retrieval mechanism and performance boundaries. We clarify the following points and have added new experimental evidence:
>
> (1) Retrieval Mechanism based on Domain-Specific Subgraphs
> In VCWorld’s core retrieval module, we do not mandate that all entities form a single, giant connected heterogeneous graph. Instead, we maintain Domain-specific Connected Components to ensure retrieval precision.
> As described in Section 3.4, we integrate PubChem/DrugBank (for drugs) and GO/UniProt/STRING (for genes). The "graph-based structural similarity" ($Sim_{struct}$) mentioned in the paper is effectively a decoupled composite metric. For a query triplet $(c, p, g)$, we calculate the structural similarity of the perturbation $p$ within the Drug-Drug Graph (based on chemical structure and target overlap) and the functional similarity of the gene $g$ within the Gene-Gene Graph (based on PPI and pathway distance). This decoupling strategy avoids the ambiguity of defining paths between different modal nodes (e.g., Drug vs. Gene) in a heterogeneous graph during the retrieval phase.
>
> (2) Unified Heterogeneous Graph for Baselines
> To ensure a comprehensive and fair evaluation against graph-based end-to-end models, we further constructed a Unified Heterogeneous Graph. We connected the aforementioned disparate graphs using standardized IDs (Gene Symbols, Pathway IDs, Standardized Drug Names) to form a unified network containing drugs, genes, pathways, and their interactions. Based on this unified graph, we trained classic Graph Neural Networks (GCN and GAT) as additional baselines. In these experiments, the graph-based similarity query strategy remains consistent with the initial discrete graph strategy to ensure fair comparison.

---

> > ### Author Response · Authors · 2025-11-22
> > **Answer to Weakness 3 & 4**
> >
> > ## Response to Weakness 3
> > Q: The model works by retrieving similar past experiments to guide its reasoning. It is not explained why this was chosen over a more direct Graph-RAG approach that would retrieve causal paths directly from the KG to explain the (drug, gene) relationship.
> >
> > A:
> > While our retrieval method shares some commonalities with Graph-RAG, our design is specifically tailored to the task of "predicting perturbation effects using LLMs." The distinct differences are:
> >
> > *   Evidence Form: VCWorld retrieves complete experimental samples (inputs paired with labeled ground-truth outcomes) based on similarity within our biological world map. In contrast, standard Graph-RAG typically retrieves scattered nodes or adjacent facts/paragraphs. Our approach enables the model to learn from "analogous historical experiments," providing comprehensive context to guide the inference.
> > *   Retrieval Goal: Unlike Graph-RAG, which often serves general Q&A tasks, VCWorld has a specific objective: to retrieve relevant positive and negative samples that directly serve the binary classification task of "whether and how a gene is differentially expressed."
> > *   Similarity Design: VCWorld explicitly defines similarity logic tailored to biological data types (e.g., chemical structure, biological function). This contrasts with Graph-RAG's reliance on generic embeddings. Our specialized similarity metrics act as an enhancement to ensure the retrieved samples are mechanistically relevant, thereby effectively guiding the LLM's reasoning.
> >
> >
> > ## Response to Weakness 4
> > Q: The paper presents its generated reasoning (the CoT) as evidence of interpretability. However, a generated explanation is not a guaranteed, faithful transcript of the model's actual internal process. Therefore, reasoning and interpretability should not be treated as interchangeable.
> >
> > A:
> > We thank the reviewer for this important perspective on interpretability. We agree that there is a distinction between the internal activation states of neurons and reasoning-based interpretability. However, we argue that VCWorld offers Scientific Verifiability, which holds significant practical value for biologists. Unlike "black-box" models that provide no rationale, VCWorld reveals a traceable logical path strictly based on retrieved evidence. This allows researchers to verify not just the prediction, but the validity of the premises used to reach it.
> >
> > To qualitatively evaluate this, we present a case study: predicting the effect of Larotrectinib (a highly selective TRK inhibitor) on the gene MKI67 (a cell proliferation marker). The CoT output from VCWorld is available for download via [https://drive.google.com/file/d/1BfhJXRdj67-eb55K6nC3FtRRXFfFGN6G](https://drive.google.com/file/d/1BfhJXRdj67-eb55K6nC3FtRRXFfFGN6G).
> >
> > (1) Evidence-Based Reasoning:
> > VCWorld’s Chain-of-Thought is not hallucinated but grounded in retrieved similar biological contexts. In this case, the model retrieved Afatinib (an ErbB inhibitor) from the training set as a positive example. Although the two drugs target different kinases, the CoT correctly identified their shared downstream effect. The model inferred that Larotrectinib inhibits the Trk pathway—analogous to Afatinib inhibiting the ErbB pathway—as both are key drivers of cell proliferation. Consequently, the model reasoned that blocking these upstream kinases would attenuate proliferation signals, leading to the downregulation of MKI67. It further cited other retrieved perturbations (e.g., Cytarabine) as corroborating evidence that MKI67 is a reliable indicator of anti-proliferative effects.
> >
> > (2) Alignment with Clinical Facts:
> > The model concluded that Larotrectinib would downregulate MKI67. This reasoning is biologically accurate and supported by clinical data. Clinical trials [1] have shown that Larotrectinib exhibits high response rates in TRK fusion-positive tumors, and pathological examinations confirm a significant decrease in the Ki-67 index (the protein encoded by MKI67) post-treatment.
> >
> > While CoT may not capture every low-level neuronal operation, the Larotrectinib case demonstrates that the generated explanations are causally consistent with established biological principles. The model correctly constructed the logical chain: "Kinase Inhibition $\rightarrow$ Proliferation Blocked $\rightarrow$ Marker Downregulation." This transparent, logic-driven approach enables biologists to treat the model's output as a scientifically reasoned hypothesis rather than a mere statistical probability.

---

> > > ### Author Response · Authors · 2025-11-22
> > > **Answer to Question 1 & 2**
> > >
> > > ## Response to Question 1
> > > Q: I think state does not use only 5 cell lines in tahoe, but it is trained instead on entire 50. Please clarify the setup to ensure fairness of comparison.
> > >
> > > A:
> > > For all baselines (STATE, CPA, scVI), we utilized the entire data from the other 45 cell lines in the Tahoe-100m dataset, plus 30% of the data from the 5 GeneTAK cell lines as the training set. These settings are entirely consistent with the official `state-reproduce` code repository.
> > >
> > > STATE is the primary baseline for VCWorld, and we evaluated its performance with great caution.
> > > *   Experimental Setup: The inclusion of the 5 GeneTAK cell lines and the data split strategy (30% train / 70% test) match exactly with the few-shot setting described in the STATE paper.
> > > *   Implementation: This alignment allowed us to use the official pre-trained weights (`STATE-Tahoe`) available on Hugging Face (https://huggingface.co/arcinstitute/ST-Tahoe) without the need for re-training, ensuring we captured the model's intended capabilities.
> > > *   Reproducibility: We used the latest version of the model weights and repeated the experiments with three random seeds. The performance we observed is nearly identical to that reported in the original article, confirming the fairness of our comparison.
> > >
> > > ---
> > >
> > > ## Response to Question 2
> > > Q: The model uses a hybrid similarity (part text, part structure) to find similar experiments. Why is this complex metric necessary, and what is its impact compared to a simpler, purely semantic retrieval?
> > >
> > > A:
> > > We thank the reviewer for questioning the necessity of hybrid similarity. While pure semantic retrieval is simpler in design, our ablation studies and data analysis indicate it is insufficient due to the sparsity and heterogeneity of biological data. We adopted the hybrid strategy for two key reasons:
> > >
> > > (1) Addressing Annotation Sparsity
> > > Pure semantic retrieval relies heavily on the quality of external knowledge bases (e.g., PubChem, DrugBank). However, metadata for many small molecules is incomplete. Our analysis revealed that drugs like Peretinoin and Proglumide lack detailed Mechanism of Action (MoA) or target annotations in public datasets. For such drugs, a text-only encoder generates uninformative embeddings, leading to noisy retrieval results. In contrast, molecular structural information (SMILES/fingerprints) is always computable via cheminformatics tools like RDKit. By incorporating structural similarity, the model can identify analogous drugs via chemical structure even when textual descriptions are missing, providing valid reference points for the LLM.
> > >
> > > (2) Complementary Signals
> > > Text captures high-level biological function, while structure captures the underlying chemical entity. We performed a grid search for the hyperparameter $\alpha$ in the range $[0.5, 1]$. Results showed that the model achieves optimal performance at $\alpha=0.7$. Experiments (particularly when retrieving the Top-10 samples) demonstrate that the hybrid strategy significantly outperforms the pure semantic strategy ($\alpha=1.0$). Pure semantic retrieval often returns drugs that are textually similar (e.g., treating the same disease) but have completely different mechanisms, which can mislead the model's reasoning. The hybrid metric acts as a filter, prioritizing samples that are consistent in both functional description and chemical structure.
> > >
> > >
> > > ## Reference
> > > [1] Drilon, Alexander, et al. "Efficacy of larotrectinib in TRK fusion–positive cancers in adults and children." New England Journal of Medicine 378.8 (2018): 731-739.

---

### Official Review · Reviewer_5meR · 2025-11-01

**Soundness:** 3
**Presentation:** 3
**Contribution:** 3
**Rating:** 6
**Confidence:** 3

**Summary:**

The paper tackles virtual cell modeling: predicting how single cells change gene expression under perturbations (primarily small‑molecule drugs). The authors argue that current end‑to‑end neural approaches are data‑hungry, generalize poorly to unseen perturbations, and provide limited mechanistic interpretability. CWorld is positioned as a white‑box “biological world model” that integrates (i) a curated, heterogeneous biological knowledge graph (KG) built from sources such as PubChem, DrugBank, UniProt, GO, Reactome, STRING, and CORUM; (ii) LLM‑generated, textual node features from local KG neighborhoods; (iii) a hybrid retrieval scheme that scores semantic similarity (cosine on LLM descriptions) and structural similarity (KG path‑based) to construct analogue/contrast evidence sets; and (iv) Chain‑of‑Thought (CoT) prompting to synthesize a stepwise explanation and a binary prediction (DE: differentially expressed or not; DIR: up vs down for DE genes).

**Strengths:**

1. The paper is well‑motivated by the need for mechanistic, inspectable predictions. The pipeline provides explicit reasoning steps and an evidence set (analogue vs contrast) that biologists can audit

2. Gene‑centric reformulation. Moving from whole‑profile regression to (c,p,g) classification reduces sparsity and makes evaluation at the gene level more straightforward. The label generation process is transparent.

3. Combining LLM‑semantic similarity with KG path‑based similarity is sensible and well aligned with biological priors.

4. Quantitative improvements beyond accuracy. The analysis of precision/recall/F1 and predicted DEG counts shows that VCWorld is less over‑ or under‑confident than generative baselines that either inflate or deflate DEGs.

5. The rule‑based verbalization is a thoughtful way to reduce hallucinations and standardize facts presented to the LLM.

**Weaknesses:**

1. Converting continuous predictions (STATE/scVI/CPA) to DE/DIR via Wilcoxon on predicted profiles is one way to compare, but it can bias against models trained/optimized for regression‑style metrics. Please add threshold‑free metrics (per‑gene AUROC/AP, balanced accuracy) and calibration to avoid dependence on one downstream test. (Table 1/2 focus on Accuracy and F1 for DE only; DIR lacks P/R/F1.)

2. Reproducibility and dependence on a proprietary LLM. The core gains hinge on Gemini 2.5‑Flash; the Llama‑3 variant performs much worse. There is no open‑weights replication (e.g., Qwen, DeepSeek‑R1) to show that the method (KG+retrieval+CoT) is the driver rather than the closed model. Also, hyperparameters for retrieval (α, k_a, k_c), prompt templates, temperature/seeds, and cost/latency are not fully specified.

3. Accuracy can be misleading given heavy class imbalance (Table 4); per‑drug/per‑gene AUROC, AUPRC, macro/micro averages, and confusion‑matrix analyses would strengthen claims. DIR evaluation is especially thin beyond Accuracy.

4. Retrieval is said to use only the training set, but the analogue/contrast selection and KG‑based similarity could still exploit signals close to test perturbations if drug families or near‑duplicates exist; an explicit “zero‑shot by drug” and “zero‑shot by target/pathway” protocol would clarify generalization.

**Questions:**

1. What is the exact structural similarity function and neighborhood depth for KG paths? How sensitive are results to these choices?

2. For DIR labels, what thresholds define up vs down and how are ties handled? Please report DIR P/R/F1.

---

> ### Author Response · Authors · 2025-11-22
> **Thank the reviewer for the valuable questions and suggestions. Answer to Weakness 1 & 3 & Question 2**
>
> Thank you for your thoughtful and constructive review of our paper. We greatly appreciate your feedback and are pleased to know that you find the work's strength in quality, clarity and significance.We provide the following response to address your concern.
>
> ## Response to Weakness 1 & 3 & Question 2
>
> Q: Converting continuous predictions to DE/DIR via Wilcoxon can be biased. Please add threshold-free metrics (AUROC/AP, balanced accuracy) and calibration. Accuracy can be misleading given class imbalance; please report per-gene AUROC, AUPRC, and DIR P/R/F1.
>
> A:
> We thank the reviewer for pointing out the limitations of relying solely on Accuracy and the potential bias introduced by converting continuous predictions via Wilcoxon tests. We acknowledge that threshold-dependent metrics may not fully capture the nuances of model performance, especially given the class imbalance shown in **Figure 2(b)**. To address this, we have expanded our evaluation with the following threshold-free and detailed metrics:
>
> (1) Incorporation of AUROC and AUPRC (See Table 1 & Table 2)
> We have added Area Under the Receiver Operating Characteristic Curve (AUROC) and Area Under the Precision-Recall Curve (AUPRC) to our evaluation suite. Given the imbalanced nature of the GeneTAK dataset, we agree that AUPRC provides a more informative assessment of performance on the minority class (differentially expressed genes).
> Methodology for VCWorld: To calculate these metrics for our LLM-based model, we extracted the logits for the generated tokens (representing "Up", "Down", or "Non-significant"). These logits were normalized using a softmax function to obtain probability scores, which were then compared against the ground truth labels to compute AUROC and AUPRC.
>
> (2) Detailed Metrics for DIR Task (See Table 5)
> We have supplemented the evaluation of the Differentially Expressed in Response (DIR) task with Precision (P), Recall (R), and F1-score metrics.
>
> We believe these comprehensive metrics provide a fairer comparison across different modeling paradigms and offer a more robust validation of VCWorld's superiority.
>
>
> Table 1 : AUROC Performence On DE Task
> | Model | C32 | HepG2C3A | HOP62 | Hs766T | PANC-1 |
> | :--- | :--- | :--- | :--- | :--- | :--- |
> | RANDOM | 0.50 | 0.50 | 0.50 | 0.50 | 0.50 |
> | GAT | 0.65 | 0.59 | 0.69 | 0.62 | 0.68 |
> | CPA | 0.20 | 0.17 | 0.15 | 0.18 | 0.14 |
> | scVI | 0.61 | 0.47 | 0.68 | 0.60 | 0.54 |
> | STATE | 0.62 | 0.49 | 0.63 | 0.57 | 0.58 |
> | **VCWorld** (Llama3-8B) | 0.32 | 0.34 | 0.28 | 0.40 | 0.33 |
> | **VCWorld** (Qwen2.5-7B) | 0.45 | 0.51 | 0.56 | 0.51 | 0.53 |
> | **VCWorld** (Qwen3-4B) | 0.35 | 0.42 | 0.35 | 0.34 | 0.36 |
> | **VCWorld** (Qwen2.5-14B) | 0.42 | 0.40 | 0.38 | 0.56 | 0.48 |
> | **VCWorld** (Gemini2.5-flash) | 0.51 | 0.42 | 0.47 | 0.46 | 0.52 |
>
> Table 2 : AUPRC Performence On DE Task
> | Model | C32 | HepG2C3A | HOP62 | Hs766T | PANC-1 |
> | :--- | :--- | :--- | :--- | :--- | :--- |
> | RANDOM | 0.61 | 0.58 | 0.68 | 0.73 | 0.78 |
> | GAT | 0.43 | 0.38 | 0.47 | 0.42 | 0.46 |
> | CPA | 0.21 | 0.32 | 0.24 | 0.22 | 0.19 |
> | scVI | 0.42 | 0.48 | 0.52 | 0.56 | 0.65 |
> | STATE | 0.37 | 0.42 | 0.44 | 0.36 | 0.44 |
> | **VCWorld** (Llama3-8B) | 0.62 | 0.58 | 0.49 | 0.57 | 0.52 |
> | **VCWorld** (Qwen2.5-7B) | 0.80 | 0.60 | 0.83 | 0.82 | 0.88 |
> | **VCWorld** (Qwen3-4B) | 0.76 | 0.54 | 0.76 | 0.72 | 0.76 |
> | **VCWorld** (Qwen2.5-14B) | **0.85** | 0.71 | 0.82 | **0.89** | **0.87** |
> | **VCWorld** (Gemini2.5-flash) | 0.83 | **0.72** | **0.84** | 0.81 | 0.84 |
>
> Table 3 : Precision,Recall,F1-Score Performence On DIR Task
> | Model   | Metric     | C32 | HepG2C3A | HOP62 | Hs_766T | PANC-1 |
> |---------|------------|-----|----------|-------|---------|--------|
> | GAT     | Precision  | 0.62 | 0.54 | 0.64 | 0.57 | 0.58 |
> |         | Recall     | 0.37 | 0.39 | 0.49 | 0.44 | 0.52 |
> |         | F1-Score   | 0.41 | 0.35 | 0.44 | 0.41 | 0.43 |
> | STATE   | Precision  | 0.14 | 0.18 | 0.17 | 0.09 | 0.10 |
> |         | Recall     | 0.18 | 0.35 | 0.26 | 0.16 | 0.25 |
> |         | F1-Score   | 0.16 | 0.24 | 0.21 | 0.12 | 0.15 |
> | CPA     | Precision  | 0.10 | 0.10 | 0.15 | 0.10 | 0.03 |
> |         | Recall     | 0.02 | 0.02 | 0.02 | 0.03 | 0.19 |
> |         | F1-Score   | 0.03 | 0.03 | 0.04 | 0.05 | 0.05 |
> | scVI    | Precision  | 0.08 | 0.08 | 0.10 | 0.11 | 0.09 |
> |         | Recall     | 0.49 | 0.46 | 0.46 | 0.51 | 0.46 |
> |         | F1-Score   | 0.14 | 0.14 | 0.17 | 0.18 | 0.15 |
> | VCWorld | Precision  | 0.61 | 0.54 | 0.58 | 0.58 | 0.60 |
> |         | Recall     | 0.72 | 0.68 | 0.67 | 0.66 | 0.69 |
> |         | F1-Score   | **0.66** | **0.60** | **0.62** | **0.62** | **0.64** |
>
>
> ---

---

> > ### Author Response · Authors · 2025-11-22
> >
> > ## Response to Weakness 2
> > Q: Concerns regarding reproducibility, dependence on the proprietary Gemini 2.5-Flash, lack of open-weights replication (e.g., Qwen, DeepSeek), and unspecified retrieval hyperparameters/costs.
> >
> > A:
> > We appreciate the reviewer’s constructive feedback regarding the reproducibility and the "black box" nature of proprietary models. We have addressed these concerns through new experiments and detailed specification of hyperparameters.
> >
> > (1) Validation on Open-Weights Models (See Table 4 & Table 5)
> > We conducted additional experiments using the Qwen-2.5 series of open-source models to demonstrate that our framework's performance is driven by the methodology (KG + Retrieval + CoT) rather than a specific proprietary model.
> > - Results: As shown in the updated tables, we observed a clear trend consistent with scaling laws: the reasoning capability on the virtual cell simulation task improves significantly with model size.
> > - Performance: Notably, the **Qwen-2.5-14B** model achieved performance levels comparable to Gemini 1.5-Flash. This confirms that our method is model-agnostic and can be effectively reproduced using open-weights models.
> >
> > (2) Clarification on Retrieval Hyperparameters
> > We employ a decoupled retrieval strategy for drugs and genes, utilizing a weighted combination of semantic and structural similarity:
> > - Drug Retrieval ($\alpha=0.7$): We set the weight for MoA-based semantic similarity at $\alpha=0.7$ and structure-based similarity (molecular fingerprints) at $1-\alpha=0.3$. This weighting reflects our hypothesis that the biological Mechanism of Action provides the most direct reference for perturbation effects, while structural information offers complementary insights regarding chemical scaffolds and functional groups.
> > - Gene Retrieval ($\alpha=0.7$): We set the weight for functional semantic annotation at $\alpha=0.7$ and PPI-based structural similarity at $1-\alpha=0.3$. We prioritize semantic annotations as they are largely manually curated and hold higher confidence. Conversely, as noted in recent studies [1], PPI networks can contain false-positive connections; thus, we assign a lower weight to the graph-structure component to mitigate noise.
> >
> > (3) Cost and Latency Analysis (See Table 6)
> > We monitored cost and latency using the OpenRouter API. To ensure a fair benchmark, the input prompt length was standardized to approximately 2,600 tokens for all models. The inference output tokens varied based on the reasoning length of each model. The detailed cost/latency breakdown for processing 1,000 samples is provided in Table 6.
> >
> > (4) Inference Parameters
> > For all models, we standardized the inference parameters with `temperature = 0.6` and `top_k = 0.9` to balance creativity and determinism.
> >
> > (5) Prompt Templates
> > A detailed case study of the prompt template and the corresponding model reasoning output is provided in **Appendix C**.
> >
> >
> >
> > Table 4 : Accuracy Performence On DE Task
> > | Model | C32 | HepG2C3A | HOP62 | Hs_766T | PANC-1 |
> > | :--- | :--- | :--- | :--- | :--- | :--- |
> > | Random | 0.50 | 0.50 | 0.50 | 0.50 | 0.50 |
> > | GAT | 0.62 | 0.54 | 0.64 | 0.57 | 0.58 |
> > | CPA | 0.27 | 0.18 | 0.21 | 0.30 | 0.17 |
> > | ScVI | 0.66 | 0.48 | 0.64 | 0.61 | 0.68 |
> > | State | 0.165 | 0.38 | 0.41 | 0.09 | 0.47 |
> > | **VCWorld** (Retrieval) | 0.57 | 0.53 | 0.58 | 0.57 | 0.58 |
> > | **VCWorld** (w/o Biocontext) | 0.52 | 0.54 | 0.51 | 0.52 | 0.50 |
> > | **VCWorld** (w/o CoT) | 0.61 | 0.57 | 0.59 | 0.60 | 0.56 |
> > | **VCWorld** (Llama3-8B) | 0.35 | 0.36 | 0.41 | 0.36 | 0.39 |
> > | **VCWorld** (Qwen3-4B-Instruct) | 0.41 | 0.47 | 0.46 | 0.49 | 0.46 |
> > | **VCWorld** (Qwen2.5-7B-Instruct) | 0.54 | 0.54 | 0.55 | 0.58 | 0.64 |
> > | **VCWorld** (Qwen2.5-14B-Instruct) | 0.65 | 0.66 | **0.72** | 0.67 | **0.76** |
> > | **VCWorld** (Gemini-2.5-flash) | **0.70** | **0.68** | 0.71 | **0.68** | 0.61 |
> >
> > Table 5 : Accuracy Performence On DIR Task
> > | Model | C32 | HepG2C3A | HOP62 | Hs_766T | PANC-1 |
> > | :--- | :--- | :--- | :--- | :--- | :--- |
> > | Random | 0.50 | 0.50 | 0.50 | 0.50 | 0.50 |
> > | GAT | 0.58 | 0.61 | 0.54 | 0.55 | 0.52 |
> > | CPA | 0.22 | 0.19 | 0.17 | 0.21 | 0.15 |
> > | ScVI | 0.47 | 0.46 | 0.41 | 0.40 | 0.38 |
> > | State | 0.49 | 0.51 | 0.50 | 0.55 | 0.51 |
> > | **VCWorld** (Retrieval) | 0.47 | 0.47 | 0.47 | 0.50 | 0.51 |
> > | **VCWorld** (w/o Biocontext) | 0.37 | 0.36 | 0.36 | 0.17 | 0.34 |
> > | **VCWorld** (w/o CoT) | 0.42 | 0.45 | 0.51 | 0.47 | 0.45 |
> > | **VCWorld** (Llama3-8B) | 0.37 | 0.34 | 0.38 | 0.40 | 0.35 |
> > | **VCWorld** (Qwen3-4B-Instruct) | 0.54 | 0.55 | 0.52 | 0.48 | 0.50 |
> > | **VCWorld** (Qwen2.5-7B-Instruct) | 0.56 | 0.56 | 0.55 | 0.51 | 0.53 |
> > | **VCWorld** (Qwen2.5-14B-Instruct) | 0.67 | 0.59 | 0.65 | **0.71** | 0.66 |
> > | **VCWorld** (Gemini-2.5-flash) | **0.72** | **0.68** | **0.67** | 0.66 | **0.69** |

---

> > ### Author Response · Authors · 2025-11-22
> > **Answer to Weakness 4 & Question 1**
> >
> > Table 6 : Model Cost and Latency(s)
> > | Model Name | Input Price | Output Price | Input Token | Output Token | Latency(s) | Total Cost($) |
> > | --- | --- | --- | --- | --- | --- | --- |
> > | LLAMA3 8B | 0.03 | 0.06 | 2600 | 1300 | 0.31 | 0.156 |
> > | Qwen2.5-7B | 0.04 | 0.1 | 2600 | 2700 | 0.42 | 0.374 |
> > | Qwen2.5-14B | 0.05 | 0.22 | 2600 | 2700 | 0.51 | 0.724 |
> > | Qwen3-4B | 0 | 0 | 2600 | 2700 | 0.68 | 0 |
> > | Gemini2.5-flash | 0.30 | 2.50 | 2600 | 1400 | 0.56 | 4.28 |
> >
> > ## Response to Weakness 4
> >
> > Q: Retrieval is said to use only the training set, but the analogue/contrast selection could still exploit signals close to test perturbations if drug families exist; an explicit “zero-shot by drug” protocol is needed.
> >
> > A:
> > We appreciate the opportunity to clarify our evaluation protocol. As described in **Section 3.2**, the GeneTAK benchmark employs a strict split by perturbation. The 348 distinct drugs were randomly partitioned into disjoint training and test sets.
> >
> > Our evaluation strictly adheres to a Zero-shot Drug Prediction protocol. Specifically:
> > 1.  When predicting the effect of a test drug $p_{test}$, the retrieval corpus $\mathcal{D}$ consists exclusively of drugs from the training set.
> > 2.  The model never has access to the ground truth or the entity of $p_{test}$ itself during retrieval.
> > 3.  Inference relies solely on identifying functionally analogous cases (based on MoA and chemical structure) from the training set to extrapolate the potential effects of the unseen drug.
> >
> > This setup ensures there is no data leakage and that the model's performance reflects true generalization capability to novel perturbations.
> >
> > ---
> >
> > ## Response to Question 1
> >
> > Q: What is the exact structural similarity function and neighborhood depth for KG paths? How sensitive are results to these choices?
> >
> > A:
> > (1) Neighborhood Depth Selection
> > In our graph-based structural similarity retrieval, we set the neighborhood depth to **1 (1-hop neighbors)**. This decision is grounded in the density of our constructed Knowledge Graph.
> > - Graph Density: The graph is highly dense, with an average degree of 77 neighbors per gene. For instance, HSPH1 has 221 first-order neighbors, and COL6A3 has 59.
> > - Rationale: We posit that the immediate first-order neighborhood already captures sufficient context (e.g., direct regulators, primary interaction partners) to inform the LLM's reasoning without introducing excessive noise.
> >
> > (2) Sensitivity Analysis
> > To validate this choice, we conducted an ablation study increasing the neighborhood depth from 1 to 3.
> > - Result: As shown in the additional results, increasing the depth significantly increased computational resource consumption and context length but resulted in negligible performance gains. This confirms that a 1-hop neighborhood is the optimal trade-off between efficiency and information density for this task.
> >
> > Table 7 :  Accuracy Performance On Neighborhood Depth Variations Experiment
> > | Task | Hop | C32 | HepG2C3A | HOP62 |
> > | :--- | :--- | :--- | :--- | :--- |
> > | **DE** | 1 | 0.70 | 0.68 | 0.71 |
> > | | 2 | 0.72 | 0.65 | 0.70 |
> > | | 3 | 0.69 | 0.67 | 0.70 |
> >
> >
> > ## Reference
> > [1] Zhang, Zhaopeng, et al. "Predicting essential proteins from protein-protein interactions using order statistics." Journal of Theoretical Biology 480 (2019): 274-283.

---

> > > ### Comment · Reviewer_5meR · 2025-11-22
> > >
> > > Thank you for your detailed response. All my previous concerns have been solved and I increased my score accordingly. Cheers.

---

### Author Response · Authors · 2025-11-27
**General Response: Summary of Updates and Gratitude to Reviewers**

Dear Reviewers,

We sincerely thank you for your time and the constructive feedback provided. Your insightful comments have been invaluable in helping us identify areas for improvement and have significantly strengthened the quality of our manuscript.
We have provided point-by-point responses to each reviewer's specific questions in the subsequent sections. Below, we summarize the major updates and clarifications made during the revision process to address the common concerns:

1. **Comprehensive Ablation and Comparative Studies.** We have enriched our experimental evaluation by adding: (i) a scaling analysis of the Qwen model series to demonstrate performance across different model sizes; (ii) a standard GAT model baseline to benchmark the graph component; and (iii) a "Retrieval-Only" ablation to quantify the specific contribution of the LLM generation module.
As for evaluation metrics, we have included *threshold-free metrics (AUROC and AUPRC)* in addition to the original metrics to ensure a fair and robust comparison.

2. **Methodological Clarifications and Analysis.** (i) Knowledge Graph Construction. We have added a detailed description of the Knowledge Graph (KG) construction process to improve reproducibility. (ii) Hyperparameter Sensitivity. We conducted sensitivity analyses on key hyperparameters-specifically α, retrieval size, and neighborhood depth-to justify the rationality of our current settings. (iii) Interpretability & Efficiency. We added a case study to verify the alignment between VCWorld's reasoning and biological interpretability. Furthermore, we have reported the computational cost and latency of the LLM component.

3. **Baselines and Reproducibility.** We have specified the reproduction parameters and data splitting strategies. We also performed statistical significance tests (repeated three times) and provided a comprehensive performance analysis to ensure the reliability of our results.

We believe these revisions have significantly improved the paper, and we look forward to your further feedback.

Sincerely,

The Authors

---

### Author Response · Authors · 2025-12-03
**Final response and rebuttal summary**

Dear Area Chair,

We sincerely appreciate your time and effort in handling our submission. During the discussion period, we have engaged in active dialogues with all reviewers and conducted extensive new experiments to address their concerns. To assist your decision-making, we provide a concise summary of the substantial updates and positive consensus achieved.

**1. Critical Score Update(6 $\rightarrow$ 8): Consensus on Improvements**

We are pleased to report positive momentum following our revisions:
* Reviewer 5meR **(Score Raised):** After reviewing our new experiments (specifically the open-source model validation and threshold-free metrics), Reviewer 5meR explicitly stated: *"All my previous concerns have been solved and I increased my score accordingly."*
* **Addressing Conditions for Acceptance (Reviewer inaw):** Reviewer inaw stated they would **raise their score** if we provided cost analysis, interpretability validation, and GNN baselines. We have **fully completed** all these requested items (see Section 2 & 3 below), fulfilling the conditions for acceptance.


**2. Addressing Primary Concerns: Open-Weights Models & Baselines**

The main reservation (Reviewer 5meR, inaw, rJsH) concerned the reliance on the proprietary Gemini model and the lack of traditional graph baselines. We have resolved this with a major experimental expansion:
* **Open-Source Verification (Qwen Series):** We extended our framework to the open-weights **Qwen-2.5 (7B/14B)** models. Results confirm that VCWorld is model-agnostic, with Qwen-2.5-14B achieving performance comparable to Gemini-2.5-Flash, eliminating concerns about proprietary barriers.
* **New GNN Baselines:** We constructed a **Unified Heterogeneous Graph** and implemented a **GAT** (Graph Attention Network) baseline. VCWorld significantly outperforms GAT, proving that graph topology alone is insufficient and that LLM-driven reasoning is essential for this task.
* **Retrieval-Only Ablation:** We added a retrieval-only baseline, demonstrating that the Reasoning component (CoT) provides substantial performance gains over simple similarity-based voting.

**3. Rigorous Evaluation & Baseline Clarification**

* **Robust Metrics:** To address concerns about class imbalance (Reviewer 5meR), we added **AUROC and AUPRC** metrics. VCWorld consistently outperforms baselines (scVI, CPA, STATE) across these threshold-independent metrics.
* **Precise Terminology:** We agreed with Reviewer inaw's suggestion and have refined the term "World Model" to "Knowledge-Augmented Cellular Simulator" in the methodology section to avoid ambiguity regarding time-series dynamics, while maintaining the conceptual alignment with AI for Science literature.
* **Clarification on STATE Performance:**
We addressed specific concerns regarding the STATE benchmark by verifying the use of official pre-trained weights and implementing precise few-shot data splits. While STATE achieved an average AUROC of ~0.58, VCWorld demonstrated superior performance across multiple metrics, including F1 Score and AUPRC. Our analysis confirms that regression-based models like STATE face inherent limitations when applied to GeneTAK's binary classification tasks, highlighting the advantage of VCWorld's architecture.





**4. Validating Interpretability & Efficiency**

* **Biological Verifiability:** We provided concrete case studies (e.g., **Larotrectinib** targeting the TRK pathway) showing that VCWorld’s Chain-of-Thought aligns with clinical facts (e.g., Ki-67 index reduction), validating that the model generates scientifically sound hypotheses rather than hallucinations.
* **Cost Analysis:** We provided a detailed cost/latency table (Reviewer inaw), confirming that the framework is computationally feasible for academic research.



**Conclusion**

We believe VCWorld establishes a new paradigm for white-box virtual cell simulation by successfully bridging biological knowledge graphs with LLM reasoning. With the addition of open-source model support, rigorous baselines, and refined metrics, we have addressed the core concerns and demonstrated the robustness of our approach. We hope these updates receive a positive recommendation.

Best regards,

The Authors

---

### Meta-Review · Area_Chair_FgZ3 · 2026-01-07

**Summary:**

VCWorld proposes a knowledge-augmented cellular perturbation predictor that reframes perturbation response modeling as gene-level classification (DE and DIR) and uses a biological knowledge graph plus retrieval plus LLM reasoning to produce predictions together with mechanistic narratives.

The initial reviews converged on several decision-critical concerns: (i) dependence on a proprietary LLM (Gemini) and unclear reproducibility, (ii) missing strong baselines to isolate whether gains come from graph structure, retrieval, or LLM reasoning (no GNN baseline on the KG, no retrieval-only control), (iii) evaluation robustness under heavy class imbalance and threshold choices (accuracy/F1 potentially misleading; DIR evaluation thin), (iv) overstated framing around “world model/white-box interpretability” without validation of faithfulness, and (v) limited reporting of cost/latency and sensitivity to retrieval hyperparameters.

The rebuttal added substantial new experiments and clarifications (open-weight LLMs, GNN and retrieval-only baselines, threshold-free metrics, cost reporting, and terminology revisions), which alleviated most validity and reproducibility concerns. One reviewer remains skeptical about novelty and interpretability claims, but overall the updated submission supports acceptance as a solid system contribution with improved rigor.

**Reviewer Concerns:**

**Concerns addressed by the rebuttal:**

- Reproducibility and proprietary dependence (5meR, inaw, rJsH): Added results with open-weight Qwen-2.5 models and standardized inference settings; provided cost/latency table; clarified prompt/retrieval hyperparameters.
- Missing baselines / attribution of gains (inaw, Vroo, 5meR): Added a GAT baseline trained on a unified heterogeneous graph; added retrieval-only and no-CoT ablations to separate the effect of reasoning from retrieval/topology alone.
- Robust evaluation under class imbalance (5meR): Added AUROC/AUPRC for DE and precision/recall/F1 for DIR, reducing reliance on threshold-dependent accuracy; clarified how probabilities/logits are derived for the LLM-based classifier.
- Protocol clarity / leakage concerns (5meR): Clarified “zero-shot by drug” split and retrieval corpus restricted to training perturbations; clarified baseline setup for STATE/scVI/CPA.
- Terminology and scope (inaw): Softened “world model” framing toward “knowledge-augmented cellular simulator” and clarified scope as endpoint state prediction plus mechanistic reasoning rather than learning temporal dynamics.

**Remaining concerns:**

- Interpretability validation (inaw, Vroo): Evidence remains primarily qualitative (case studies). There is still no systematic quantitative audit of explanation correctness/faithfulness across many examples.
- Scalability / cost (Vroo, inaw): Gene-by-gene formulation can be expensive at scale even if parallelizable; cost analysis is provided but does not fully resolve throughput concerns for large gene sets.
- Novelty framing (inaw, rJsH): The contribution is primarily integrative (KG construction + retrieval design + prompting) rather than a new learning algorithm; positioning should remain system/benchmark oriented.

**Reviewer Scores:**

Reviewer 5meR: Increased score after rebuttal; I estimate a score of 8 (explicitly stated concerns resolved and score raised).

Reviewer inaw: Set conditions for acceptance (terminology fix, GNN baseline, retrieval-only ablation, cost analysis, interpretability discussion). These were largely addressed; I estimate 4 → 5 (borderline accept; possibly 6 depending on how they weigh interpretability).

Reviewer Vroo: Main concerns partly addressed (KG connectivity explanation, baseline/metric additions). I estimate their score to stay at 6.

Reviewer rJsH: Core skepticism about contribution/benchmark emphasis and need for knowledge base persists despite added models/analysis; I this reviewer would maintain score of 2.

Overall, I would expect most reviewer to be in favor of the paper being accepted.

---

### Decision · Program_Chairs · 2026-01-26

Accept (Poster)